# BLINK-Twice: You see, but do you observe?
# A Reasoning Benchmark on Visual Perception

**Junyan Ye[1,2], Dongzhi Jiang[3], Jun He[1], Baichuan Zhou[2], Zilong Huang[1],**
**Zhiyuan Yan[4], Hongsheng Li[3], Conghui He[2], Weijia Li[1, †]**
[1]Sun Yat-sen University, [2]Shanghai Artificial Intelligence Laboratory,
[3]CUHK MMLab, [4]Peking University
[†] Corresponding author: `liweij29@mail.sysu.edu.cn`

## Abstract

Recently, Multimodal Large Language Models (MLLMs) have made rapid progress, particularly in enhancing their reasoning capabilities. However, existing reasoning benchmarks still primarily assess language-based reasoning, often treating visual input as replaceable context. To address this gap, we introduce BLINK-Twice, a vision-centric reasoning benchmark grounded in challenging perceptual tasks. Instead of relying on external knowledge, our tasks require models to reason from visual content alone, shifting the focus from language-based to image-grounded reasoning. Compared to prior perception benchmarks, it moves beyond shallow perception ("see") and requires fine-grained observation and analytical reasoning ("observe"). BLINK-Twice integrates three core components: seven types of visual challenges for testing visual reasoning, natural adversarial image pairs that enforce reliance on visual content, and annotated reasoning chains for fine-grained evaluation of the reasoning process rather than final answers alone. We evaluate 20 leading MLLMs, including 12 foundation models and 8 reasoning-enhanced models. BLINK-Twice poses a significant challenge to current models. While existing reasoning strategies in the language space—such as chain-of-thought or self-criticism can improve performance, they often result in unstable and redundant reasoning. We observe that repeated image observation improves performance across models, and active visual interaction, as demonstrated by models like o3, highlights the need for a new paradigm for vision reasoning. The dataset is publicly available at `https://github.com/PicoTrex/BLINK-Twice`.

## 1 Introduction

> *"You see, but you do not observe."* — *Sherlock Holmes*

In recent years, Multimodal Large Language Models (MLLMs) built upon Large Language Models (LLMs) and advanced vision encoders have rapidly progressed. Both closed models like GPT-4o and Gemini, and open-source systems such as LLaVA, InternVL, and QwenVL, have demonstrated impressive visual perception, even surpassing human performance on certain tasks [36, 35, 50, 28, 63, 27, 52]. With the emergence of reasoning-augmented models such as OpenAI's o1 [37] and Deepseek-R1 [15], which leverage chain-of-thought (CoT) [47] and reinforcement learning techniques, reasoning has become a growing focus. Notably, this shift extends beyond language reasoning into the multimodal domain, as evidenced by the strong reasoning capabilities of models like Visual-RFT[30] and the recently released o3 [38].

To quantify the reasoning capabilities of MLLMs, the research community has proposed various multimodal reasoning benchmarks [59, 39, 58, 5, 17]. For instance, MMMU [57] evaluates models

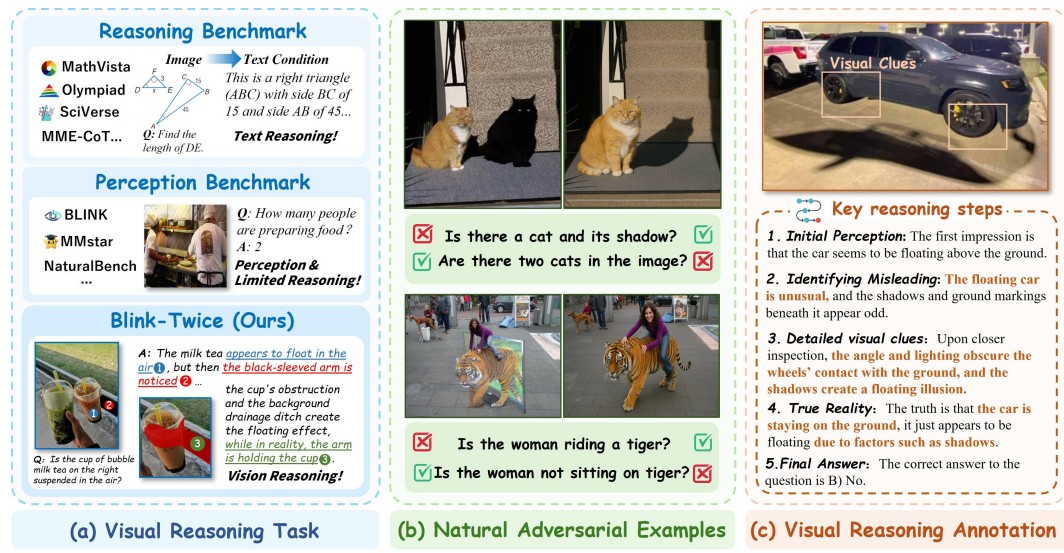

Figure 1: BLINK-Twice Task Overview: (a) Visual Reasoning task requiring detailed observation and careful reasoning; (b) Natural adversarial samples with similar appearance but opposite semantics, forcing models to rely on visual input; (c) Reasoning step annotation including detailed visual clues and true reality to evaluate thought chain output.

on college-level questions to assess knowledge-based reasoning; MathVerse [58] and Olympiad-Bench [18] focus on mathematical and physical reasoning challenges; MME-CoT [22] specifically targets chain-of-thought reasoning capabilities. However, most existing benchmarks remain centered on textual knowledge and logical reasoning, with visual input serving merely as auxiliary context—sometimes even replaceable by simple textual cues. These benchmarks rely more on the language model's knowledge and logical reasoning, while overlooking in-depth understanding and reasoning based on the visual content itself. Hence, there is an urgent need for a multimodal benchmark centered on vision-driven reasoning.

To this end, we propose the BLINK-Twice: A Reasoning Benchmark on Visual Perception. It starts from fundamental visual perception tasks, ensuring that answers depend on image content rather than prior knowledge or mathematical reasoning. By introducing more challenging visual perception and reasoning tasks, BLINK-Twice emphasizes the need to not only "see" but to truly "observe"—simple perception is no longer sufficient, and models must consciously attend to visual details and reason to fully comprehend the image. This aligns with the principle: "You see, but you do not observe." This turns visual perception tasks into a test of reasoning ability. This differs from previous perception benchmak such as BLINK [11] and NaturalBench [25], which primarily focus on direct perception tasks with limited reasoning requirements.

As illustrated in Figure 1, BLINK-Twice incorporates three key aspects: **i.** Our visual reasoning tasks span seven carefully curated **visual challenging** collection, such as visual dislocation, forced perspective, and motion illusion, enabling comprehensive evaluation of models' perception and reasoning capabilities; **ii.** Leveraging GPT-4o's powerful image editing capabilities, we construct **natural adversarial image pairs**—visually similar yet semantically distinct—forcing models to rely on detailed visual perception; **iii.** We provide **annotated reasoning chains and key detail scoring points**, enabling fine-grained analysis of reasoning quality and efficiency. Together, these designs establish BLINK-Twice as a strong framework for advancing the evaluation and development of multimodal reasoning systems beyond solely relying on final answer accuracy.

We systematically evaluate 20 leading MLLMs, including 12 foundation MLLMs and 8 reasoning-enhanced models incorporating chain-of-thought mechanisms. Our key findings are as follows:

- ***Current MLLMs exhibits potential flaw in visual reasoning, often "see" but fail to "observe".*** Even GPT-4o and Gemini-2.5 Pro perform suboptimally in both final answers (I-Acc, G-Acc) and reasoning processes (CoT-Score).

- ***Step-by-step reasoning or self-criticism helps reasoning models tackle complex visual reasoning challenges.*** QVQ, Claude-3.7-Thinking, and Gemini-2.0-Flash-Thinking outperform their base models by a notable margin.

- ***Reasoning models often overextend their reasoning chains, leading to redundanct reasoning.*** Efficiency analysis reveals that models like QVQ often over-criticize, producing redundant steps even after finding the correct answer.

- ***MLLM reasoning require new paradigms of visual reasoning, rather than relying solely on inference in the text space.*** Compared to single-pass perception followed by language reasoning, multi-turn dialogues with repeated image observation significantly enhance performance. The latest o3 model further demonstrates a novel paradigm through active visual reasoning via dynamic image cropping and transformation.

## 2  Related Work

**Multimodal Large Language Models and Reasoning Model.**  In recent years, multimodal large language models (MLLMs) have achieved remarkable performance across both general tasks [28, 54, 42, 64] and specialized domains [48, 24, 55, 49, 16, 32]. Notable open-source models include LLaVA [28], and Qwen-VL [42], while closed-source counterparts such as GPT-4o [36] and Gemini [12] excel in advanced visual understanding and reasoning. With the release of o1 [37], the development of large models has increasingly shifted towards enhancing reasoning capabilities. DeepSeek R1 [15] and QwQ [41] enhance reasoning performance by generating intermediate reasoning steps, or Chains of Thought (CoT)[47], before final answers. Similarly, in the domain of MLLMs, early studies [60] explored adapting the CoT reasoning paradigm to vision-language tasks such as VQA and chart interpretation. Recent methods such as MM-EUREKA [34], Visual-RFT [30], and Skywork-R1V [40] employ large-scale rule-based RL to enhance multimodal reasoning [45, 21]. Meanwhile, closed-source models like Gemini [12, 13] and Claude [1, 2] have also introduced experimental versions designed specifically to enhance reasoning. OpenAI's latest o3 [38] model further advances image-level reasoning, enabling more meticulous observation and inference through operations like cropping and rotation. As multimodal reasoning models advance, effective benchmarks are needed to better evaluate their visual reasoning capabilities.

**Multimodal & Reasoning Benchmarks.**  Meanwhile, numerous benchmarks have been proposed to evaluate the capabilities of MLLMs [29, 20, 56, 10, 9, 23, 33]. For instance, benchmarks such as BLINK [11], MM-Star [4], and NaturalBench [25] focus on assessing perception and understanding abilities through tasks like image captioning, counting, and basic spatial reasoning, effectively measuring fundamental visual recognition skills. In contrast, many benchmarks focus on evaluating the reasoning capabilities of MLLMs in domains such as expert knowledge, mathematics, and science [59, 39, 58, 5, 17, 19, 6]. For example, MMMU [57] uses college-level questions to assess models' mastery of expert knowledge and complex reasoning. MathVista [31] and Olympiad-Bench [18] further provide challenging problems in mathematics and physics to evaluate logical reasoning capabilities.

Compared to prior benchmarks focused on visual perception [11, 4, 25, 61, 10], BLINK-Twice moves beyond "See and Get" tasks by emphasizing reasoning and interpretation of image semantics. Unlike MathVista [31] and OlympiadBench [18], which focus on mathematical or scientific reasoning, our task design centers on visual reasoning. Most existing reasoning benchmarks treat visual input as auxiliary context for assisting reasoning occurring in the language domain. In some cases, visual input can even be replaced by textual cues, suggesting that it is not essential for reasoning. In contrast, BLINK-Twice highlights multimodal reasoning driven by image content. It also incorporates natural adversarial samples that compel the model to perform in-depth image analysis, along with detailed reasoning chain annotations to evaluate the quality, stability, and efficiency of reasoning—beyond mere answer accuracy. Additional dataset comparisons are in the supplementary materials.

## 3  Dataset

We introduce BLINK-Twice, a benchmark designed to evaluate models' visual reasoning capabilities. It contains 345 challenging base images across 7 types of visual challenges. These images were initially collected from over 650 samples across multiple internet platforms. Due to the benchmark's

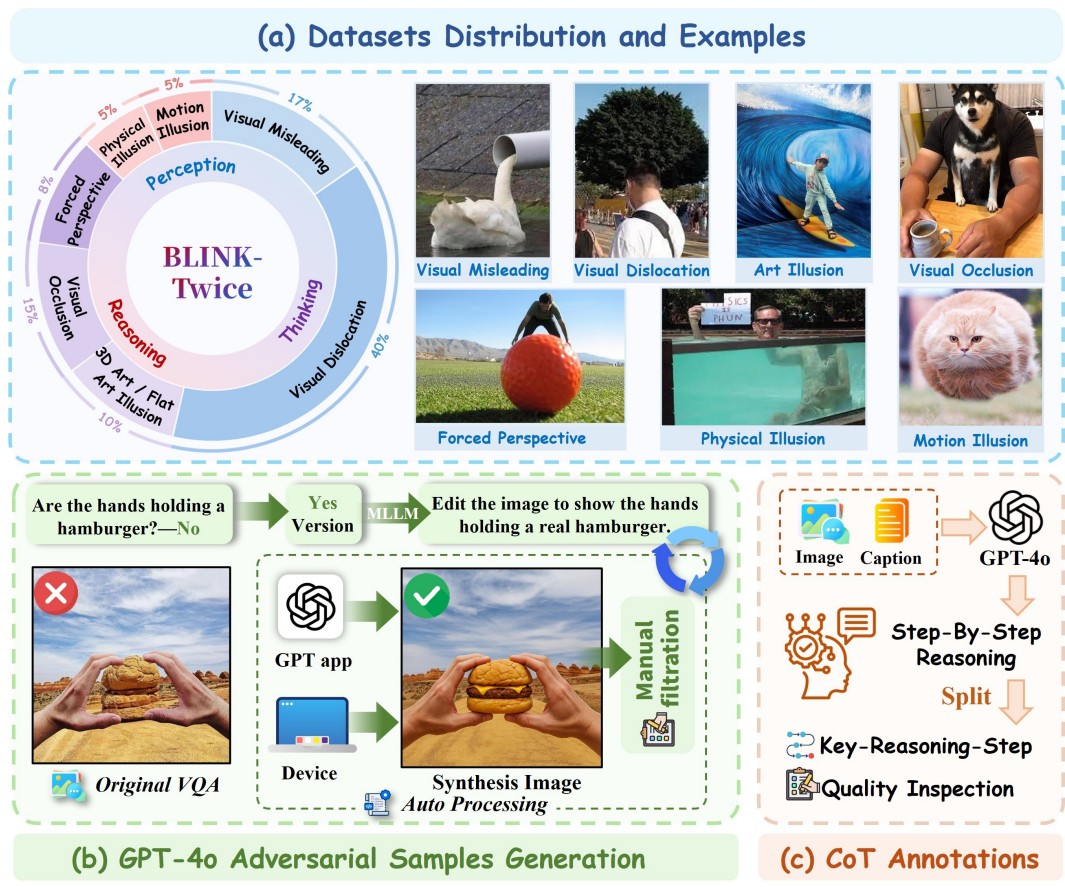

Figure 2: Overview of BLINK-Twice Dataset. (a) Distribution and examples of different visual challenges; (b) Pipeline for automatic adversarial sample generation and (c) reasoning chain annotation.

high requirements on visual ambiguity, scene diversity, and reasoning complexity, the data collection and filtering process was particularly demanding. Ultimately, only images that truly serve reasoning evaluation purposes were retained. Collection sources are detailed in the supplementary materials.

Additionally, leveraging the powerful image-editing capability of GPT-4o [36], we produce 103 natural adversarial samples. These samples are manually curated to ensure they are visually similar yet fundamentally different in factual content. To assess MLLMs' performance, we have curated 896 manually crafted VQA questions. Furthermore, the dataset includes 1,725 annotated reasoning step, generated through GPT-4o [36] and human-constructed prompts, highlighting two critical scoring aspects: detailed visual cues and true reality.

## 3.1 Visual Challenge Classification

BLINK-Twice comprises real-world, visually challenging images collected from the internet, categorized into seven fine-grained types, each representing distinct origins and mechanisms of visual misperception. Figure 2(a) shows the dataset distribution, with a brief description of each category provided below:

**Visual Misleading:** Errors in object recognition caused by coincidental alignments of color, shape, or composition — e.g., a swan's head in a pipe mistaken for a water splash due to color similarity.

**Visual Dislocation:** Spatial coincidence between foreground and background elements creates positional ambiguity — e.g., a man standing in front of a tree appears to have an exaggerated "afro" due to the alignment of the foliage with his head.

**Art Illusion:** Flat paintings or landscape art simulate three-dimensional effects — e.g., a boy standing on a painted surface appears to be surfing on ocean waves.

**Visual Occlusion:** Partial occlusion leads to identity or structural misjudgment — e.g., a dog blocking a man's face gives the illusion that the man is wearing a "dog head mask."

**Forced Perspective:** Manipulated camera angles and depth cues create size distortions — e.g., a golf ball close to the lens looks enormous, while a distant man appears to be holding it.

**Physical Illusion:** Natural physical phenomena such as reflection, refraction, or lighting distort visual interpretation — e.g., water refraction makes a man's head appear detached from his body.

**Motion Illusion:** Static images capture high-speed movement, creating a false sense of motion — e.g., a mid-air kitten appears to be floating due to its dynamic pose frozen at the peak of a leap.

## 3.2 Adversarial Samples Generation

To encourage image-grounded reasoning, we constructed natural adversarial samples, where each question pairs visually similar but semantically contrasting images with opposite answers, reducing reliance on commonsense and enhancing visual understanding. This setup discourages shortcut learning based on commonsense priors and compels the model to engage in genuine visual understanding. Unlike prior methods that rely on semi-automated CLIP-based filtering [25], we leverage the state-of-the-art image editing capabilities of GPT-4o to generate semantically altered yet locally consistent adversarial examples. This method not only simplifies data collection but also enables fine-grained and controllable semantic modifications tailored to specific reasoning challenges.

As illustrated in Fig. 2(b), we start with an original VQA sample (e.g., "Are the hands holding a hamburger? — No") and aim to generate a factual image yielding the opposite answer "Yes". We then utilize MLLM to interpret the image and question, producing structured editing instructions (e.g., region, object type, reference). For instance, "Edit the image to show the hands holding a real hamburger." Finally, GPT-4o is employed to perform image editing and synthesize the natural adversarial sample. Notably, since OpenAI's official image editing API was not available at the time, we used an automated scripting approach [51] to batch process and retrieve the edited images. Due to the strict image generation policies of OpenAI, we obtained only 243 initial samples, which were further manually filtered to retain only those meeting our quality and specification requirements.

## 3.3 Reasoning-step Annotation & Review

As shown in Fig. 2(c), we illustrate the reasoning annotation process. We start with human-labeled image facts (e.g., "The image looks like [A], but is actually [B]"), where [A] denotes the misleading appearance and [B] the ground truth, and then generate corresponding questions. GPT-4o is then guided to produce a step-by-step reasoning chain across five stages: Initial Perception, Identifying Misleading, Detailed Visual Clues, True Reality, and Final Answer. The "Detailed Visual Clues" and "True Reality" steps are defined as key reasoning steps and used to evaluate CoT scores. Finally, the annotated reasoning chains are manually reviewed for quality.

To mitigate potential hallucinations in GPT-4o's reasoning path annotations, we performed human validation for all samples in this task. Each response involving GPT was reviewed in at least two independent rounds. A total of five annotators participated in the verification process, with an additional 60 hours spent on this phase. In addition, all generated natural adversarial samples and their corresponding VQA questions are manually verified, taking approximately 50 hours in total.

# 4 Experiments

In this section, we evaluate a range of MLLMs, including both open and closed-source models, on our proposed benchmark. All evaluations are conducted under a zero-shot setting. We begin by introducing the evaluated models and our evaluation protocol. We then analyze the performance of existing MLLMs on challenging visual reasoning tasks, with a particular focus on their reasoning capabilities under our task configuration. Finally, we discuss potential directions toward advancing multimodal reasoning in future work.

Table 1: Evaluation results of open-and closed-source MLLM across multiple metrics. ☆ indicates CoT-enhanced variants; **bold** indicates the best result, and underlined denotes the second best.

| Model | No-Acc | Yes-Acc | Q-Acc | I-Acc | G-Acc | CoT Score |
|---|---|---|---|---|---|---|
| *Open-source MLLMs* | | | | | | |
| InternVL2-8B [8] | 0.367 | 0.596 | 0.478 | 0.194 | 0.083 | 0.194 |
| InternVL2-26B [8] | 0.529 | 0.325 | 0.429 | 0.188 | 0.120 | 0.288 |
| InternVL2-40B [8] | 0.514 | 0.466 | 0.491 | 0.276 | 0.140 | 0.301 |
| InternVL2.5-8B [7] | 0.350 | 0.582 | 0.463 | 0.199 | 0.099 | 0.287 |
| MM-Eureka-8B [34] ☆ | 0.319 | 0.610 | 0.461 | 0.176 | 0.078 | 0.285 |
| Qwen2.5-VL-7B [53] | 0.410 | 0.543 | 0.475 | 0.262 | 0.078 | 0.340 |
| MM-Eureka-Qwen-7B [34] ☆ | 0.452 | 0.507 | 0.479 | 0.265 | 0.109 | 0.339 |
| Qwen2-VL-72B [52] | 0.372 | 0.614 | 0.491 | 0.233 | 0.061 | 0.341 |
| QVQ-72B [45] ☆ | 0.517 | **0.637** | 0.575 | 0.336 | 0.067 | **0.438** |
| Qwen-2.5-VL-32B [53] ☆ | 0.631 | 0.523 | **0.578** | **0.353** | **0.158** | 0.328 |
| Qwen-2.5-VL-72B [53] | **0.653** | 0.380 | 0.520 | 0.261 | 0.152 | 0.360 |
| *Closed-source MLLMs* | | | | | | |
| Claude-3.5-sonnet [1] | 0.693 | 0.282 | 0.496 | 0.190 | 0.076 | 0.539 |
| Claude-3.7-sonnet [2] | 0.680 | 0.134 | 0.414 | 0.085 | 0.035 | 0.526 |
| Claude-3.7-sonnet-thinking [2] ☆ | 0.717 | 0.274 | 0.502 | 0.189 | 0.101 | 0.536 |
| Gemini-1.5-flash [44] | 0.410 | 0.591 | 0.499 | 0.250 | 0.130 | 0.365 |
| Gemini-2.0-flash [13] | 0.360 | 0.694 | 0.525 | 0.242 | 0.071 | 0.469 |
| Gemini-2.0-flash-thinking [13] ☆ | 0.503 | 0.583 | 0.542 | 0.353 | 0.156 | 0.470 |
| GPT-4o [36] | 0.616 | 0.523 | 0.571 | 0.351 | 0.198 | **0.601** |
| o1 [37] ☆ | 0.710 | 0.503 | 0.608 | 0.392 | 0.186 | – |
| Gemini-2.5-pro [14] ☆ | **0.729** | **0.600** | **0.667** | **0.470** | **0.269** | 0.584 |

## 4.1 Experimental Setup

**Evaluated Models:** For open-source models, we evaluate representative and strong MLLMs such as InternVL [8, 7, 3] and Qwen series [52, 53, 45], as well as the reasoning-oriented MM-EUREKA [34] model built upon them. For closed-source models, we include advanced commercial systems such as Claude [1, 2], Gemini [12, 13, 14], and the GPT [36, 37] series. In total, we evaluate 20 high-performing MLLM, including 8 models with dedicated CoT reasoning optimization. A complete list of evaluated models is provided in the supplementary material.

**Evaluation Protocols:** We evaluate model performance using accuracy on our constructed VQA tasks. Each image is typically associated with two binary questions: a main question (answered "no") and an adversarial one (answered "yes"), corresponding to No-Acc and Yes-Acc metrics, respectively. To comprehensively assess model behavior, we follow NaturalBench [25] to use Q-Acc (either question correct), I-Acc (both questions per image correct), and G-Acc (all four questions in a group correct) metrics. Additionally, inspired by the reasoning steps evaluation of large models [22, 46], we propose a CoT-score to assess reasoning chain quality, based on annotated reasoning steps and GPT-4o scoring. The score is grounded on two key points: identifying detailed visual cues (1 point) and inferring the true reality (1 point). Multiple valid reasoning paths are allowed; direct yet logically sound answers receive the full 2 points as well. The final score is normalized to the [0,1] range. Most reasoning models produce directly evaluable chains, while typical MLLMs require step-by-step prompting to elicit such reasoning. Further evaluation details are in the supplementary materials.

## 4.2 Challenge to MLLMs

As shown in Table 1, BLINK-Twice poses a significant challenge to current multimodal models. Among open-source models, reasoning-enhanced approaches such as Qwen-2.5-VL-32B and QVQ achieve the strongest results. Notably, the smaller 32B model, after reinforcement learning-based reasoning enhancement, performs comparably to or even surpasses the earlier 72B version. Although QVQ is built on the previous Qwen2-VL architecture, it still achieves competitive results due to its self-criticism mechanism. In the InternVL series, performance consistently improves with larger language models despite using the same InternViT-6B vision encoder, indicating the positive impact of model scale on visual understanding. Among proprietary models, Gemini-2.5 Pro and OpenAI

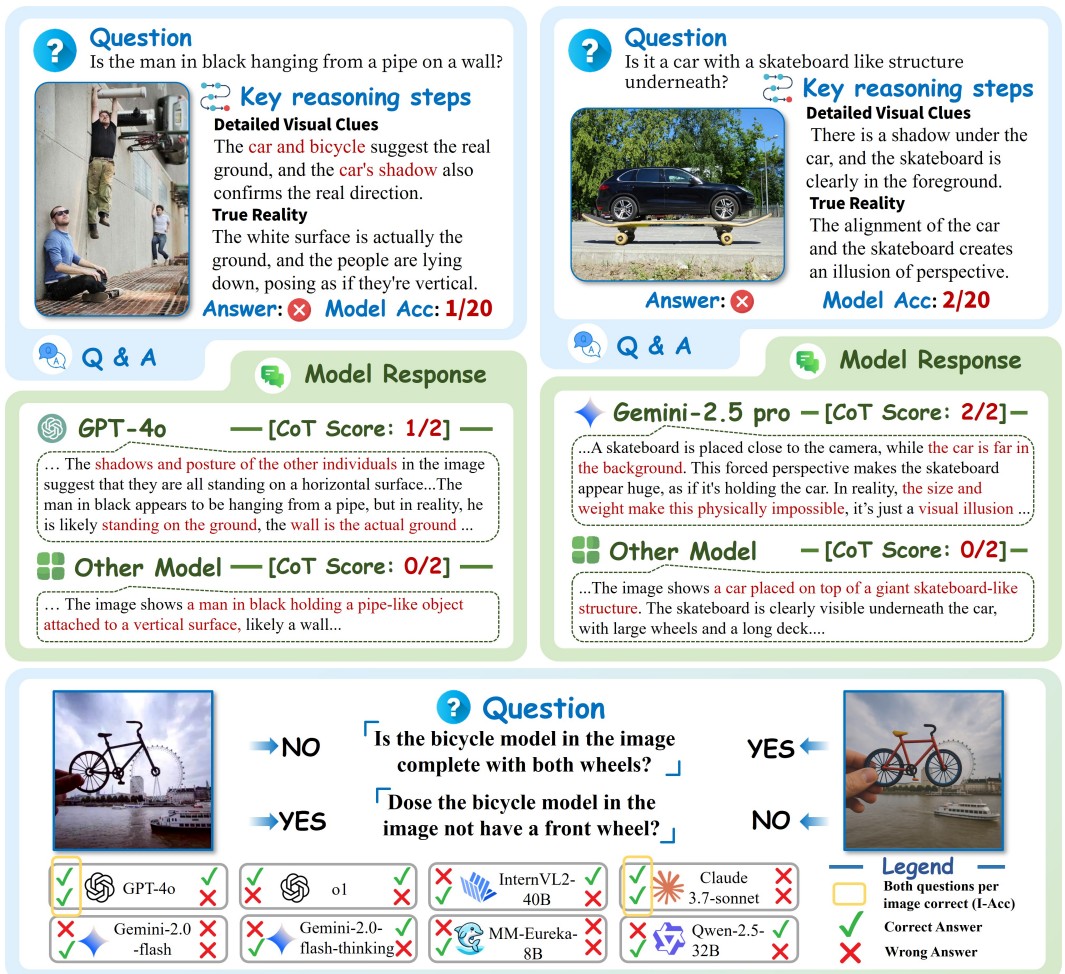

Figure 3: BLINK-Twice qualitative evaluation visualization results. The upper part shows the results of a single difficult image. The lower part displays the results for a group of images.

o1 perform best, with the former slightly outperforming the latter. Nevertheless, performance on more challenging metrics such as I-Acc and G-Acc remains suboptimal, with I-Acc below 0.5 and G-Acc under 0.3, highlighting the ongoing limitations of current multimodal systems in complex visual reasoning tasks. Current models often remain at the level of visual perception, lacking genuine understanding and reasoning over images.

Compared to answer-level accuracy (No-Acc), the CoT-score tends to be lower, suggesting that some correct answers may result from guesses rather than genuinely sound reasoning. Among open-source models, QVQ demonstrates relatively high-quality reasoning. Similarly, GPT-4o exhibits strong reasoning performance. Figure 3 presents visualized results from model testing. Only a few models, such as GPT-4o and Gemini-2.5 Pro, answer certain questions correctly, while others consistently fail. Even correct answers may overlook key visual cues—for example, missing distant vehicles that imply the true ground in the first question—resulting in a CoT Score of 1/2 rather than full credit. The CoT Score focuses on evaluating the reasoning process rather than the final answer alone. In the adversarial QA examples below, only GPT-4o and Claude-3.7 answered two questions correctly for the same image, and no model succeeded on all four. This highlights the stringent demands that I-Acc and Q-Acc place on both visual perception and reasoning.

## 4.3 Reasoning Models Analysis

As shown in Figure 4 (a), we compare the performance of various base MLLMs and reasoning-augmented models on BLINK-Twice. The results indicate that incorporating chain-of-thought reasoning significantly enhances performance on this visual reasoning benchmark. For instance,

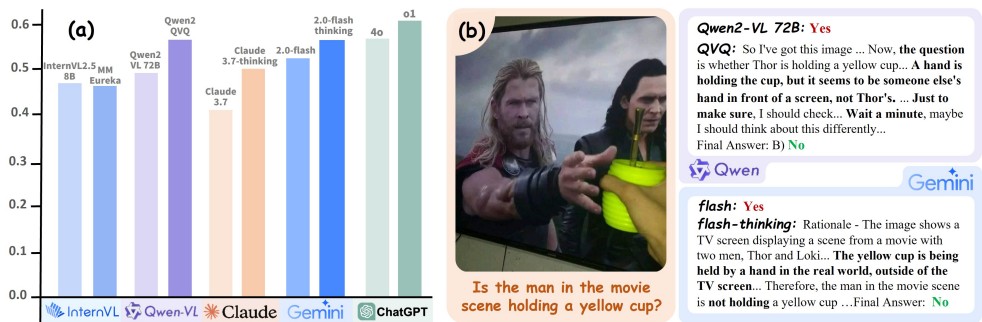

Figure 4: Comparison and Visualization of Reasoning Models and MLLMs.(a) Comparison of Q-Acc between reasoning models and based MLLMs. (b) The visualization illustrates the model's response process, highlighting step-by-step reasoning and self-critical chains of thought.

QVQ outperforms QwenVL2-72B by 15%, and the Claude-3.7 Thinking (16k) variant achieves a 20% improvement over its non-thinking counterpart. Similarly, Gemini-2.0-Thinking surpasses Gemini-2.0-Flash, and o1 also shows notable gains over GPT-4o. In contrast, MM-Eureka exhibits minimal improvement over InternVL2.5-8B, likely because its fine-tuning focuses primarily on mathematical reasoning, contributing less to general visual reasoning capabilities.

In Figure 4(b), we provide a qualitative visualization of reasoning processes. Compared to the base Qwen2-VL, the QVQ model performs step-by-step reasoning over the image and question, even without explicit prompts. For example, it first identifies the key visual cue: "A hand is holding the cup, but it seems to be someone else's hand in front of a screen, not Thor's.". Then, using phrases like "Just to make sure" or "Wait a minute," the model engages in self-refutation and reflective reasoning, ultimately providing a clear answer. Similarly, Gemini-2.0-Flash-Thinking also demonstrates a structured reasoning path to arrive at the correct answer.

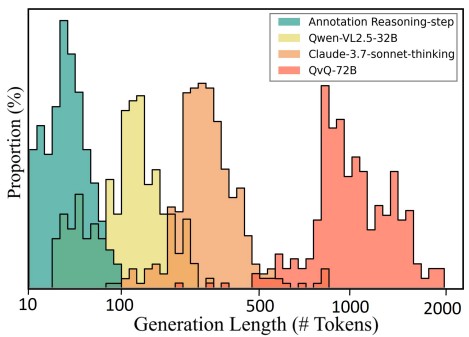

Figure 5: The distribution of generation length of Reasoning model.

While reasoning-augmented models achieve notable performance gains, such improvements often come at the cost of efficiency. As shown in Figure 5, we compare the length of annotated standard reasoning steps with the actual outputs from different models. The standard reasoning length generally remains below 100 tokens. Qwen-2.5-VL-32B, trained with chain-of-thought supervision, maintains efficient reasoning with an average output length of around 120 tokens. Claude-3.7-sonnet-thinking responds more slowly, with moderate efficiency. In contrast, QVQ consistently generates lengthy outputs—over 950 tokens on average—regardless of question difficulty or answer certainty, relying heavily on self-contradiction processes. While this improves accuracy, it also introduces significant redundancy. Future reasoning models should pursue more adaptive, selective strategies, akin to human reasoning, focusing effort only when necessary rather than indiscriminately extending the reasoning chain.

## 4.4 Multimodal Reasoning Paradigms

In prior reasoning model paradigms, the visual modality primarily serves for perception and feature extraction, while reasoning is predominantly driven by the language modality [26]. In such setups, the image is often encoded only once at the input stage—following a "see once" strategy, such as global feature extraction using CLIP [43]—and subsequent integration and logical inference rely entirely on the language model. However, for visual reasoning tasks, relying solely on language-based generation of intermediate reasoning steps may be limiting; repeatedly perceiving the image during reasoning can enhance the reliability of model decisions. To evaluate this, we design a multi-turn dialogue setup in which the model views the image twice before answering. As shown in Figure 6(a), models with weaker initial visual capabilities—such as Gemini-2.0-flash-thinking and Qwen2VL-72B exhibit notable performance improvements. In contrast, models with already strong visual grounding, such as

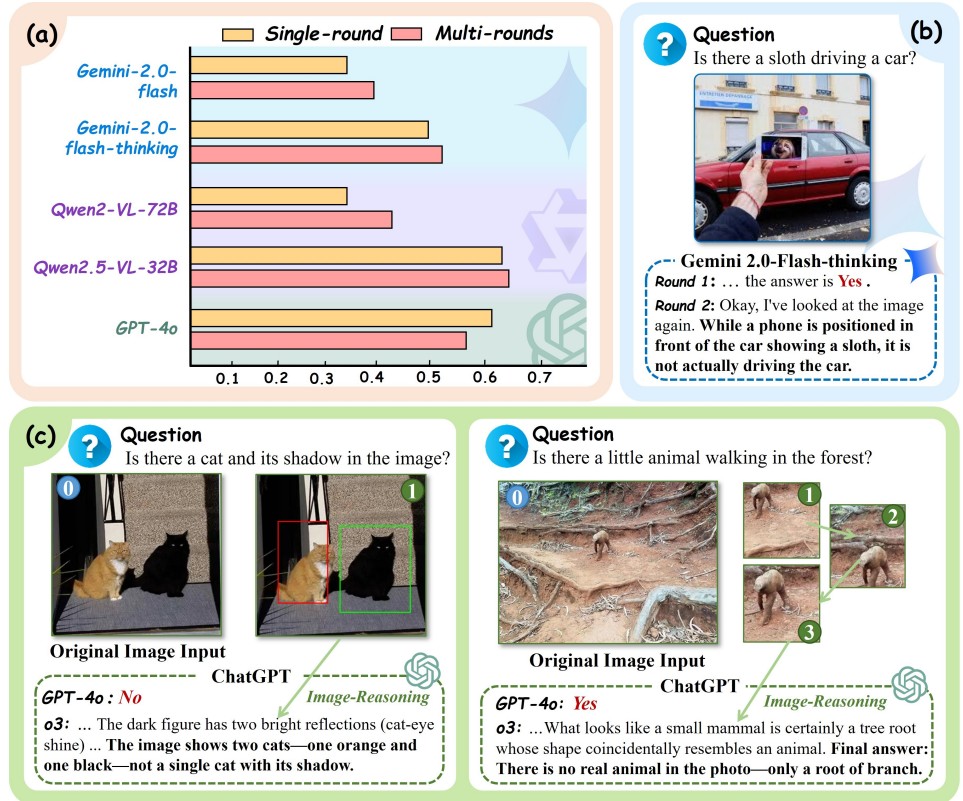

Figure 6: Multi-turn dialogue and multimodal reasoning. (a) shows the model's performance in single/multi-turn dialogue settings, (b) visualizes Gemini results in multi-turn dialogue, and (c) demonstrates the latest o3 model's ability to perform multimodal reasoning using image tools.

GPT-4o and QwenVL2.5-72B, show limited further gains. In Figure 6(b), Gemini-2.0-flash-thinking correctly identifies a raised phone only after a second observation.

As reasoning models continue to evolve, the visual modality should move beyond its passive role of perception and instead engage in collaborative reasoning with the language modality. In this paradigm, the visual module not only responds to language instructions but also initiates internal reasoning actions, such as invoking visual tools for image editing or transformation [62, 26], thereby forming explicit reasoning trajectories within the visual feature space. The recently released o3 model from OpenAI exhibits signs of such an evolution. As illustrated in Figure 6(c), the o3's reasoning process involves operations like "generating auxiliary bounding boxes" or "progressively zooming into image regions" as part of its internal deliberation. This suggests a promising trajectory for multimodal reasoning architectures, and underscores the value of our proposed BLINK-Twice dataset as a challenging and diagnostic benchmark for evaluating such capabilities. Further analysis and visualizations of multimodal reasoning are provided in the supplementary materials.

## 5 Conclusion

We introduce BLINK-Twice, a benchmark designed to systematically evaluate the performance of current multimodal large language models on complex visual perception and reasoning tasks. Experimental results indicate that current models exhibit notable limitations in visual perception and reasoning, often focusing on surface-level perception rather than truly understanding image content through reasoning. While Chain of Thought reasoning can partially improve performance on visual reasoning tasks, language-centric reasoning models remain dependent on the initial perception results and struggle with efficiency and redundancy issues. Future reasoning approaches are likely to evolve toward fully multimodal paradigms, such as repeatedly querying image perception or actively interacting with images for more robust visual reasoning. We hope BLINK-Twice will serve as a valuable benchmark for advancing research on perception and reasoning, and foster continued development in multimodal reasoning.

## Acknowledgement

This work was supported by the National Natural Science Foundation of China (Grant No. 62571560) and Shanghai Artificial Intelligence Laboratory.

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
