# Supplementary Material for
# BLINK-Twice: You see, but do you observe?
# A Reasoning Benchmark on Visual Perception

This supplementary material provides additional details omitted from the main paper due to space limitations. It includes a more comprehensive description of the dataset (Section A), covering data collection, comparisons with existing datasets, and additional visualizations. We also present extended experimental details (Section B), including the full list of evaluated models, the computation of evaluation metrics, analysis of multimodal reasoning paradigms, and more qualitative visual results. Finally, we discuss the limitations of our method (Section C).

## A  Supplementary Dataset Details

### A.1  Data Collection

Figure 3 illustrates our data collection pipeline. We began by gathering approximately 15,000 raw visual illusion image pairs from various online sources. However, most of these samples depict classical illusions—such as geometric, color, and brightness illusions—which are not aligned with our focus on visual reasoning in natural scenes. To address this, we employed GPT-4o [13] to automatically classify and filter the collected images, removing most classical illusion samples. This filtering step reduced the dataset to just over 600 images. We then conducted a manual quality inspection to ensure visual clarity, reasoning feasibility, and balanced coverage across different perceptual challenge types. The crowdsourcing process complied with local regulations and compensation standards. As a result, we curated a final dataset of 345 high-quality image pairs for BLINK-Twice.

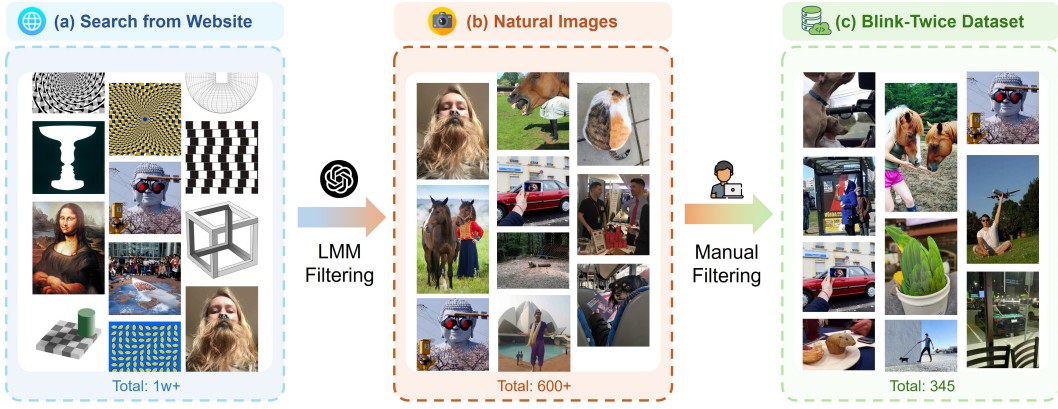

Figure 1: Data collection and filtering process.

The raw image samples were collected from the following publicly available online sources. Among these, we would like to especially thank the Bilibili user[1] whose contributions played a significant role in enriching our dataset with diverse visual illusions.

- `https://pixabay.com/images/search/optical%20illusion/`
- `https://www.msn.com/en-us/lifestyle/travel/24-weird-pics-of-optical -illusions-in-real-life/ss-BB1nzpZN#image=3`
- `https://www.boredpanda.com/funny-optical-illusions/`
- `https://space.bilibili.com/3546772500646401`

---

[1] `https://space.bilibili.com/3546772500646401`

Submitted to 39th Conference on Neural Information Processing Systems (NeurIPS 2025). Do not distribute.

- `https://www.gettyimages.co.jp/search/2/image?phrase=nature+optical+i` `llusion`
- `https://www.istockphoto.com/jp/search/2/image-film?page=2&phrase=op` `tical+illusion`
- `https://cheezburger.com/8918533/27-images-of-weird-perspective-tha` `t-produced-real-life-optical-illusions`
- `https://www.pinterest.com/pin/natural-optical-illusion--51812513210` `8101056/`
- `https://www.shutterstock.com/zh/search/natural-illusions?dd_referrer` `=https%3A%2F%2Fwww.google.com%2F`
- `https://stock.adobe.com/search?k=illusion+nature`

## A.2  Dataset comparison

To further distinguish BLINK-Twice from existing benchmarks, we present a detailed comparison in Figure 2. The left side of the figure illustrates a two-dimensional distribution of datasets. Most reasoning-focused benchmarks are concentrated along the vertical axis, emphasizing deep reasoning in domains such as mathematics and science. However, these datasets place limited demands on visual perception, as images often serve merely as background to support textual reasoning. In contrast, perception-centric datasets lie along the horizontal axis. They focus on tasks such as image captioning, object counting, and basic visual question answering. These benchmarks require strong perceptual capabilities but lack reasoning depth. While they challenge models in visual recognition, they do not adequately assess the integration of perception and logical reasoning. BLINK-Twice occupies a middle ground between these two axes. It is designed to evaluate models' visual reasoning capabilities by requiring both accurate perception and thoughtful inference over complex real-world images. This positions BLINK-Twice as a bridge between visual perception and reasoning benchmarks.

The right side of the figure 2 provides a structured comparison between BLINK-Twice and three categories of existing benchmarks: Illusion, Vision, and Reasoning. Compared to datasets such as GVIL and IllusionVQA, which focus on classical optical illusions (e.g., geometric, brightness, and depth illusions), BLINK-Twice emphasizes perceptual challenges arising in real-world scenarios. Unlike hallucination-centered studies like HALLUSION Bench—which highlight errors caused by language model pretraining biases—BLINK-Twice focuses on errors stemming from limitations in visual perception and understanding, not merely linguistic hallucinations. Moreover, BLINK-Twice incorporates natural adversarial examples that require models to engage in fine-grained visual analysis. Each sample includes annotated reasoning chains, enabling comprehensive evaluation of model performance in terms of reasoning quality, stability, and efficiency—not just final answer.

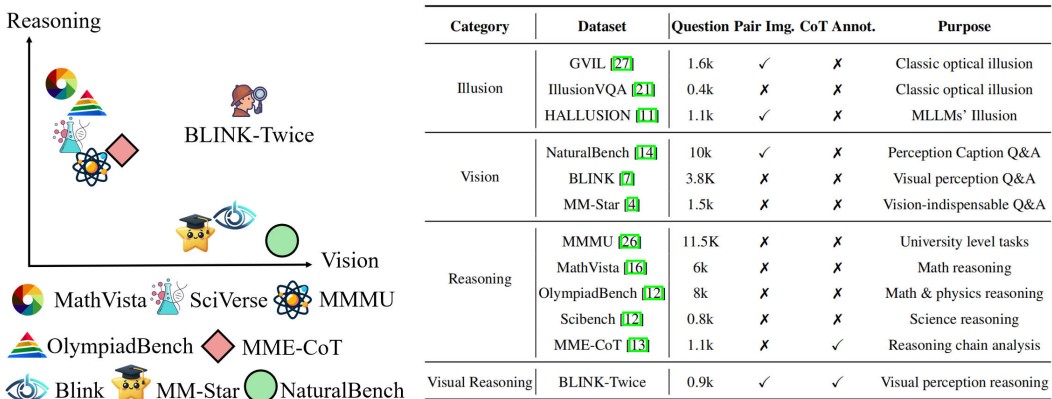

| Category | Dataset | Question Pair | Img. | CoT Annot. | Purpose |
|----------|---------|---------------|------|------------|---------|
| Illusion | GVIL [27] | 1.6k | ✓ | ✗ | Classic optical illusion |
|  | IllusionVQA [21] | 0.4k | ✗ | ✗ | Classic optical illusion |
|  | HALLUSION [11] | 1.1k | ✓ | ✗ | MLLMs' Illusion |
| Vision | NaturalBench [14] | 10k | ✓ | ✗ | Perception Caption Q&A |
|  | BLINK [7] | 3.8K | ✗ | ✗ | Visual perception Q&A |
|  | MM-Star [4] | 1.5k | ✗ | ✗ | Vision-indispensable Q&A |
| Reasoning | MMMU [26] | 11.5K | ✗ | ✗ | University level tasks |
|  | MathVista [16] | 6k | ✗ | ✗ | Math reasoning |
|  | OlympiadBench [12] | 8k | ✗ | ✗ | Math & physics reasoning |
|  | Scibench [12] | 0.8k | ✗ | ✗ | Science reasoning |
|  | MME-CoT [13] | 1.1k | ✗ | ✓ | Reasoning chain analysis |
| Visual Reasoning | BLINK-Twice | 0.9k | ✓ | ✓ | Visual perception reasoning |

Figure 2: Comparison between BLINK-Twice and existing multimodal benchmarks. The left plot illustrates the distribution of benchmarks along two dimensions: visual perception demand and reasoning depth. The table on the right highlights key differences between BLINK-Twice and representative datasets from the Illusion, Vision, and Reasoning categories.

## A.3 More data visualizations

Figure 3 presents representative examples from the BLINK-Twice dataset. Images labeled as "1" correspond to the original visual challenge samples, while those labeled as "2" are natural adversarial counterparts generated from the originals. The accompanying text below each image pair highlights key scoring points that models should focus on during reasoning, including critical visual cues and the underlying semantic truth of the scene.

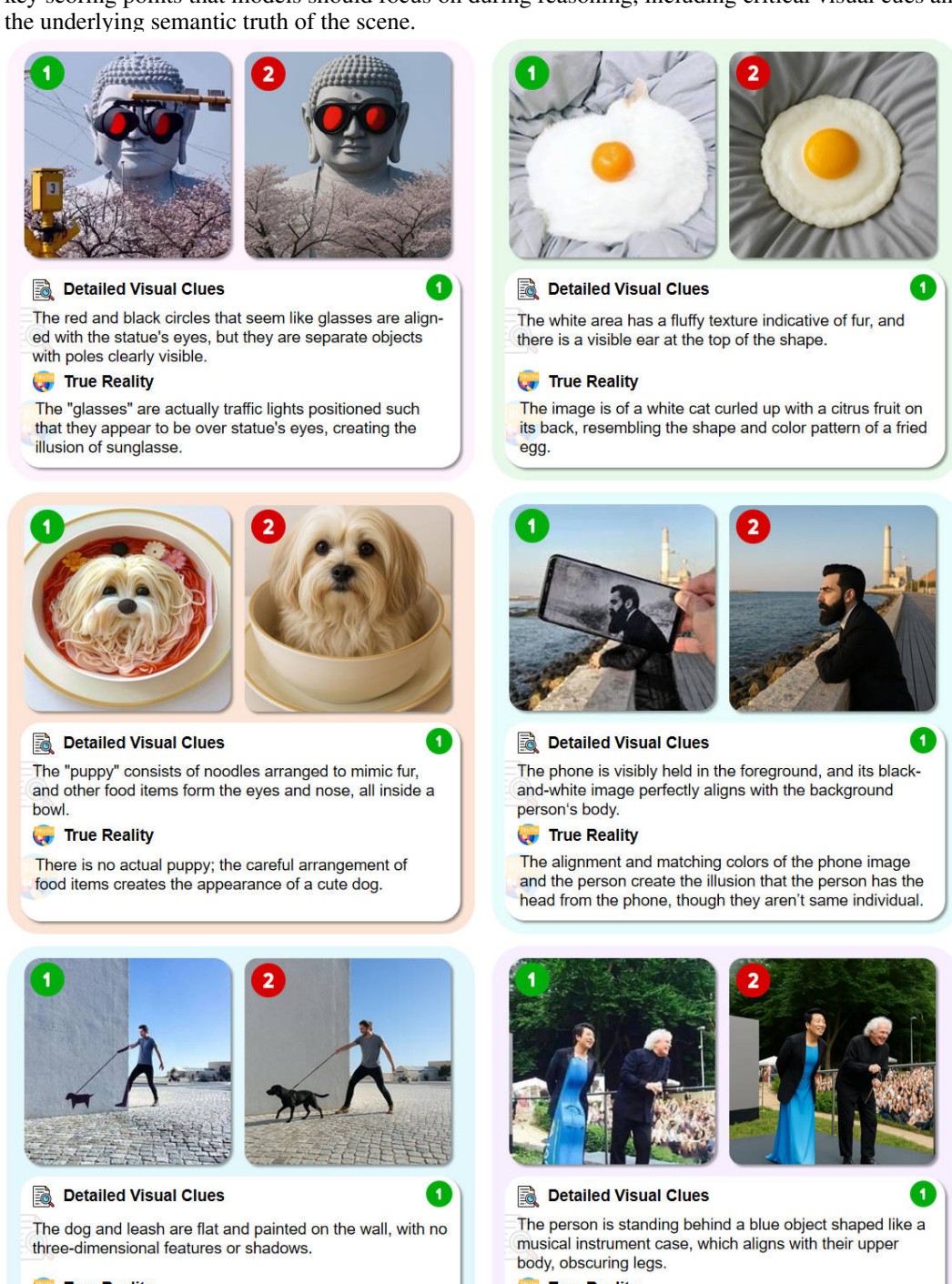

Figure 3: Representative Examples and Key Reasoning Clues in the BLINK-Twice Dataset.

## B    Additional Experimental Analysis

### B.1    Evaluation Model

We benchmark a broad range of models on the BLINK-Twice dataset to assess their capabilities in visual reasoning tasks. Our evaluation includes 12 foundational multimodal models and 8 reasoning-enhanced models with chain-of-thought capabilities. For open-source models, we select representative and high-performing MLLMs such as the InternVL series [5, 4, 3], the Qwen series [18, 19, 16], and the reasoning-focused MM-EUREKA model built upon them [12]. For closed-source models, we evaluate advanced commercial systems including the Claude series [1, 2], Gemini series [6, 7, 8], and the GPT series [13, 14]. A detailed list of the evaluated models is provided below. Proprietary models are evaluated via API calls, while open-source models are tested either through API access or locally using dual A100 GPUs. All models are the latest available versions as of April 2025.

| Model Family | Model Version | Parameters | Links |
|---|---|---|---|
| **Open-sourced** | | | |
| InternVL | InternVL2-8B | 8B | https://huggingface.co/OpenGVLab/InternVL2-8B |
| | InternVL2-26B | 26B | https://huggingface.co/OpenGVLab/InternVL2-26B |
| | InternVL2-40B | 40B | https://huggingface.co/OpenGVLab/InternVL2-40B |
| | InternVL2.5-8B | 8B | https://huggingface.co/OpenGVLab/InternVL2_5-8B |
| Qwen | Qwen2-VL-72B | 72B | https://huggingface.co/Qwen/Qwen2-VL-72B-Instruct |
| | QVQ | 72B | https://huggingface.co/Qwen/QVQ-72B-Preview |
| | Qwen2.5-VL-7B | 40B | https://huggingface.co/Qwen/Qwen2.5-VL-7B-Instruct |
| | Qwen2.5-VL-32B | 40B | https://huggingface.co/Qwen/Qwen2.5-VL-32B-Instruct |
| | Qwen2.5-VL-72B | 8B | https://huggingface.co/Qwen/Qwen2.5-VL-72B-Instruct |
| MM-Eureka | MM-Eureka-8B | 8B | https://huggingface.co/FanqingM/MM-Eureka-8B |
| | MM-Eureka-Qwen-7B | 7B | https://huggingface.co/FanqingM/MM-Eureka-Qwen-7B |
| **Closed-sourced** | | | |
| Gemini | Gemini-1.5-Flash | N/A | https://ai.google.dev/gemini-api/docs/models/gemini#gemini-1.5-flash |
| | Gemini-2.0-flash | N/A | https://ai.google.dev/gemini-api/docs/models/gemini#Gemini-2.0-flash |
| | Gemini-2.0-flash-thinking | N/A | https://ai.google.dev/gemini-api/docs/models/gemini#gemini-2.0-flash-thinking-exp-1219 |
| Claude | Claude-3.5-Sonnet | N/A | https://docs.anthropic.com/en/docs/about-claude/models |
| | Claude-3.7-Sonnet | N/A | hhttps://docs.anthropic.com/en/docs/about-claude/models |
| OpenAI | GPT-4o | N/A | https://platform.openai.com/docs/models/gpt-4o |
| | o1 | N/A | https://platform.openai.com/docs/models/o1 |

### B.2    Evaluation Metric

BLINK-Twice adopts a VQA-style question-answering format, and we use *accuracy* as the primary evaluation metric. The accuracy is calculated as:

$$\text{Accuracy} = \frac{N_{\text{correct}}}{N_{\text{total}}} \times 100\%$$

where $N_{\text{correct}}$ denotes the number of correctly answered questions and $N_{\text{total}}$ is the total number of questions. In addition, inspired by the NaturalBench setup [10], we introduce three hierarchical metrics for evaluating model robustness under natural adversarial settings: **Q-Acc** (either question in a pair is correct), **I-Acc** (both questions associated with an image are correct), and **G-Acc** (all four questions in a logical group are correct).

Inspired by recent work on step-by-step reasoning evaluation in large models [9, 17], we propose a **CoT Score** to assess the quality of reasoning chains, leveraging annotated reasoning steps and GPT-4o-based evaluation. Our scoring scheme is based on two key criteria: (1) *Recognition of detailed visual cues* (0–1 point), and (2) *Accurate inference of the underlying truth* (0–1 point). Multiple valid reasoning paths are allowed. A concise and logically sound direct answer that correctly

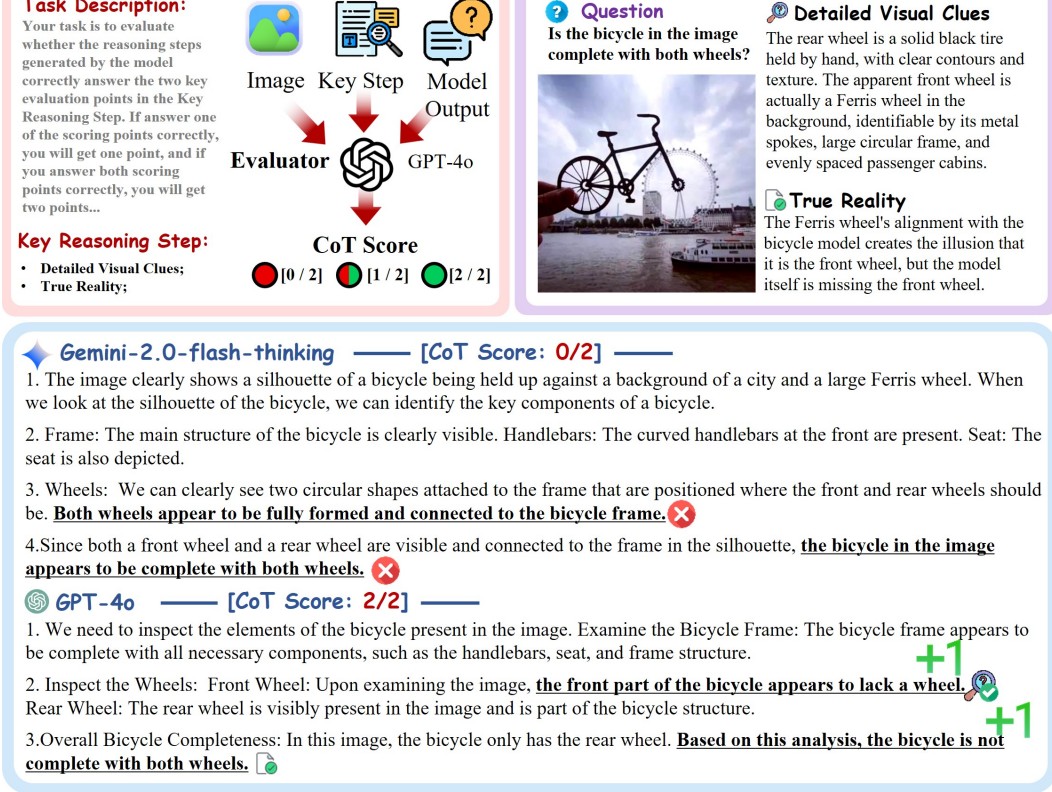

Figure 4: Example of CoT-Score Evaluation with Reasoning Outputs and Scoring Results

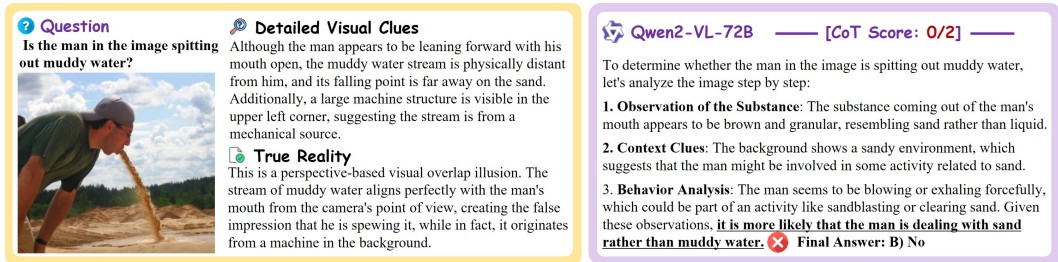

Figure 5: Correct Answer with Incomplete Reasoning: A Low CoT-Score Case

addresses the visual reasoning challenge can also receive a full score of 2. The final CoT Score is normalized to the range [0, 1]. This metric enables a fine-grained assessment of reasoning quality beyond simple answer correctness, offering deeper insights into model behavior.

Figure 4 illustrates a concrete example of our CoT-Score evaluation. The upper-left section presents the task description and required inputs for the scoring process, including the reference image and the annotated Key Reasoning Steps (Key-Steps), alongside model-generated reasoning outputs to be evaluated. The lower section displays the scoring results. Although Gemini-2.0-flash-thinking demonstrates step-by-step reasoning, it fails to detect the absence of the bicycle's front wheel—thus missing both the detailed visual clue and the underlying reality. In contrast, GPT-4o correctly identifies the missing front wheel. While it omits mention of the Ferris wheel in the background, our framework permits diverse reasoning paths, as long as they arrive at the correct factual conclusion. Figure 5 presents a case where the model provides a correct answer, yet receives a relatively low CoT-Score. This is because the reasoning overlooks key visual cues—such as the large machine in the background, the trajectory of the splashing mud, and its proximity to the person—and instead focuses on abstract semantic differences between mud and sand.

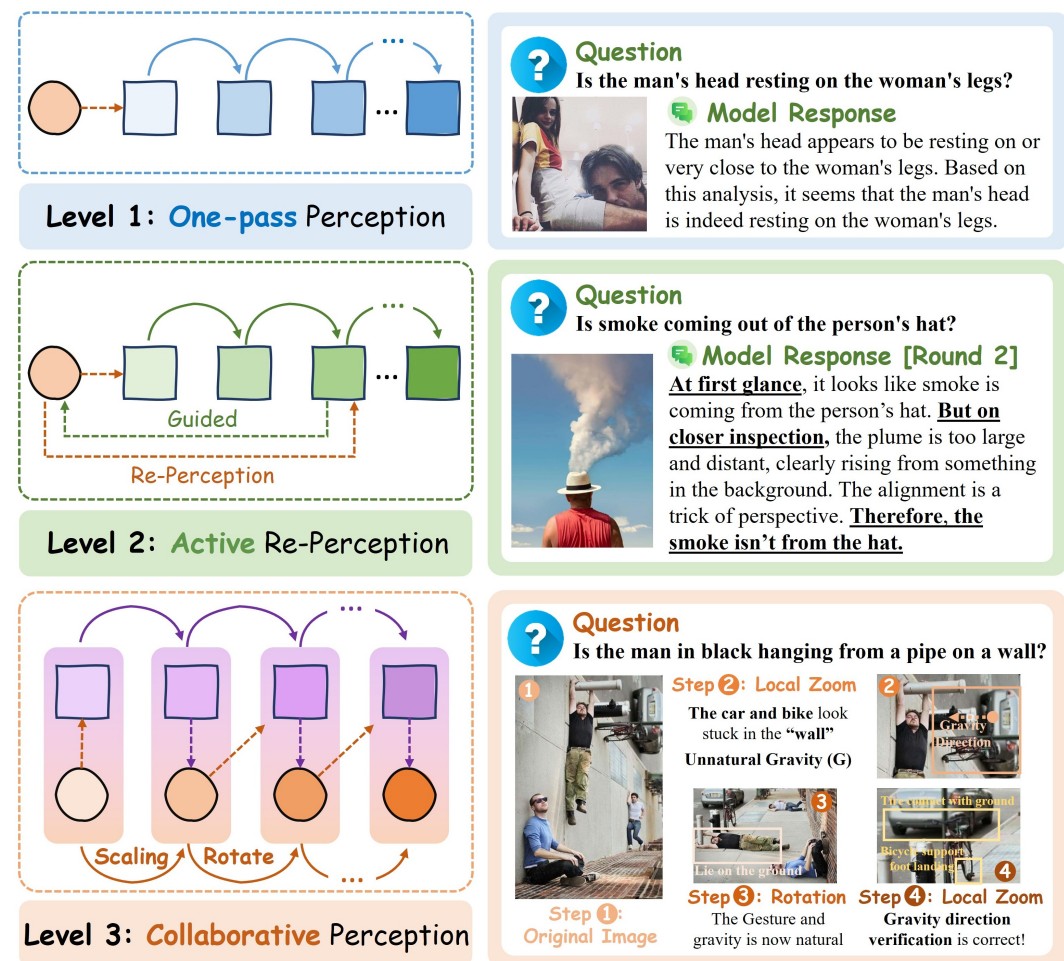

Figure 6: Illustration of Multimodal Reasoning Paradigms: One-Pass, Active Re-Perception, and Collaborative Perception. This taxonomy draws on insights from [11], which provides an excellent summary of paradigms in multimodal reasoning.

### B.3 Multimodal Reasoning Paradigm

As illustrated in Figure 6, current paradigms for multimodal reasoning can be broadly categorized into three types:

**(a) One-Pass Perception Reasoning:** In this setting, the image is encoded only once at the input stage using a visual backbone such as CLIP [15], and all subsequent reasoning is performed purely within the language modality [11]. The visual component serves primarily as a perception module, while the reasoning chain is entirely language-driven. However, for visual reasoning tasks, relying solely on language-based intermediate reasoning often proves insufficient.

**(b) Active Re-Perception Reasoning:** As shown in Figure 6(b), models re-access the image based on textual feedback during the reasoning process. In our multi-turn dialogue setup, for example, the model is explicitly prompted to revisit the image before answering. This enforced re-attention notably improves performance in models with weaker initial visual capabilities (e.g., Gemini-2.0-flash-thinking and Qwen2VL-72B), enhancing their perceptual grounding and reasoning reliability.

**(c) Collaborative Perception Reasoning:** In this emerging paradigm (Figure 6(c)), the vision module actively participates in reasoning by executing internal visual operations—such as rotation or local magnification [20, 11]—rather than passively responding to text prompts. The recently introduced o3 model by OpenAI demonstrates early signs of this shift, integrating visual interaction as part of its reasoning pipeline.

The evaluation results on BLINK-Twice reveal clear challenges for models that follow the first paradigm, highlighting the need for more active and perceptually grounded visual reasoning. Meanwhile, BLINK-Twice serves as a comprehensive and systematic benchmark to assess and guide future developments in multimodal reasoning.

**B.4 More experimental results**

We further include the detailed metric results corresponding to Fig 4 and 6, which are presented as bar charts in the main text. Table 1 demonstrates the advantages of our reasoning model over various MLLMs on visual reasoning tasks. Table 2 highlights that under the multi-turn dialogue paradigm, models achieve more effective final decisions by re-observing the image across multiple rounds.

Table 1: Quantitative Comparison of single-turn / multi-turn dialogue settings.

| | Model | Q-Acc |
|---|---|---|
| **Single-round** | Qwen2-VL-72B | 0.372 |
| | Gemni-2.0-flash | 0.360 |
| | Qwen-2.5-VL-32B | **0.631** |
| | Gemni-2.0-flash-thinking | 0.503 |
| | GPT-4o | 0.616 |
| **Multi-round** | Qwen2-VL-72B | 0.452 |
| | Gemni-2.0-flash | 0.401 |
| | Gemini-2.0-flash-thinking | 0.527 |
| | Qwen-2.5-VL-32B | **0.638** |
| | GPT-4o | 0.565 |

Table 2: Quantitative Comparison of Reasoning Models and MLLMs.

| Model | Q-Acc |
|---|---|
| InternVL2.5-8B | **0.463** |
| MM-Eureka-8B ☆ | 0.461 |
| Qwen2-VL-72B | 0.491 |
| QVQ-72B ☆ | **0.575** |
| Claude-3.7-sonnet | 0.414 |
| Claude-3.7-sonnet-thinking ☆ | **0.502** |
| Gemini-2.0-flash | 0.525 |
| Gemini-2.0-flash-thinking ☆ | **0.542** |
| GPT-4o | 0.571 |
| o1 ☆ | **0.608** |

**B.5 Case Study**

Figures 7–9 provide an in-depth comparative analysis of GPT-4o, Qwen-VL-72B, QVQ, and Gemini-2.0-thinking on visual reasoning tasks, with particular attention to erroneous predictions. This analysis is crucial for understanding each model's reasoning capabilities and limitations. It not only identifies current weaknesses but also offers insights for future improvements in model design.

# C Discussion of Limitations

While BLINK-Twice marks a promising step forward in evaluating multimodal reasoning, particularly visual reasoning, it is important to acknowledge several limitations.

The current categorization of questions is based on the source of visual challenges. Nonetheless, incorporating difficulty-level distinctions could provide a more fine-grained understanding of how MLLMs address problems of varying complexity. This added dimension may offer deeper insights into model generalization and reasoning robustness, which we leave for future work.

BLINK-Twice currently focuses on image-based reasoning tasks. As model architectures and training strategies continue to evolve, future MLLMs are expected to demonstrate enhanced reasoning capabilities in temporally dynamic contexts such as video. Expanding BLINK-Twice to support video-based visual reasoning would therefore be a meaningful direction.

Moreover, questions in BLINK-Twice and most existing benchmarks are predominantly in English, which limits the evaluation of multilingual MLLMs. While visual content remains constant, reasoning chains may vary across languages due to differences in linguistic structure, cultural context, and semantic interpretation. Incorporating multilingual visual reasoning tasks would not only increase the dataset's global relevance but also enable a more comprehensive assessment of MLLMs' linguistic diversity and cross-lingual reasoning ability.

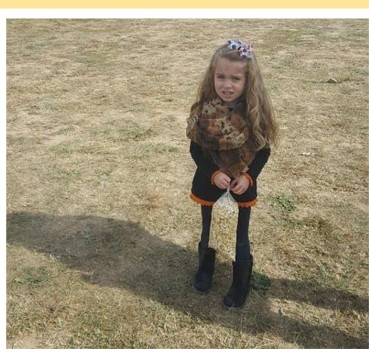

**❓ Question**

**Do the little girl in the image have very thin legs?**

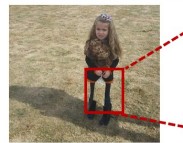

**🔍 Detailed Visual Clues**

The lower part of the girl's legs is obscured by a bag of popcorn that is the same color as the grass, making her legs appear unusually thin.

**📄 True Reality**

The bag of popcorn blends with the ground, covering part of her legs and creating the illusion that they are much thinner than they actually are.

---

**GPT-4o**: To determine whether the little girl in the image has very thin legs, I'll go through the following steps: 1. **Examine the Proportions**: Look at the overall size and proportion of the legs relative to the rest of the body, such as the torso and head. 2. **Consider Optical Illusions**: Sometimes, the angle of the photograph or the child's pose can create optical illusions, making the legs appear thinner than they are in reality. 3. **Analyze Clothing**: The type of clothing worn can affect the perception of leg thickness. Dark or tight-fitting clothing can sometimes make legs appear thinner. Upon reviewing the image: The girl's legs appear unusually thin compared to the rest of her body. The position and angle of the photo may contribute to this perception.- She is wearing dark clothing, which can enhance the illusion of thinness. **Final Answer: A) Yes**

---

**Qwen2-VL-72B**: To answer this question, let's analyze the image step by step: 1. **Identify the subject**: The image shows a young girl standing outdoors. 2. **Focus on the legs**: The girl is wearing black tights and black boots. The tights are form-fitting, which allows us to see the shape of her legs. 3. **Assess leg thickness**: From the visible portion of her legs, they appear to be proportionate for a child of her age. There are no signs of excessive thinness or unusual proportions. 4. **Consider context**: Children's body proportions can vary, but the girl's legs do not appear unusually thin compared to typical childhood development. Given this analysis, the girl's legs do not appear to be very thin. **Final Answer: B) No**

---

**Gemni-2.0-flash-thinking**: **Rationale**: The question asks if the little girl in the image has "very thin legs". To answer this, we need to visually assess the girl's legs in the image. Looking at the image, especially the cropped versions focusing on her legs, we can observe the following: 1. **Slenderness**: Her legs are noticeably slender, especially in comparison to the size of her boots and the rest of her body. 2. **Proportionality (to some extent)**: While thin, her legs don't appear completely disproportionate to her upper body at first glance. However, they are definitely on the thinner side. 3. **Clothing**: She is wearing dark tights or leggings, which can sometimes make legs appear thinner than they are. However, even considering this, her legs still appear quite thin. Considering the term "very thin", it's subjective. However, in the context of the image, her legs are clearly not thick or average in thickness for a child. They are visibly slender and could be described as "thin" or even "very thin" without being an exaggeration. If we compare her leg thickness to what might be considered average or thicker legs for a child her age, they would fall on the thinner end of the spectrum. Therefore, given the choice between "Yes" and "No", and based on visual interpretation, "Yes" is the more appropriate and accurate answer to the question. While "very thin" might be a slightly strong descriptor, it's not inaccurate and certainly more fitting than "No". **Final Answer: A**

**QVQ**: So I've got this image of a little girl standing in what looks like a park or an open field. The first thing I notice is that she's dressed in a scarf and long sleeves, which suggests that it might be a bit chilly outside. Her scarf has a pattern on it, maybe something like leaves or flowers, and she's wearing black leggings with orange accents around the knees. She also has on black boots that come up to her calves. Her hair is long and blonde, and it's tied back with a bow that has pink and white colors. She's standing with her hands on her hips, which gives her a confident or playful pose. Behind her, there's a parking lot filled with cars, indicating that there might be an event going on nearby or that it's a popular spot. The field itself has some patches of green grass, but it's mostly dry and brown, which could mean it's either late autumn or early spring. There are trees in the background with yellowing leaves, further supporting the idea that it's fall. The sky is clear and blue, so it's a sunny day despite the cooler weather. Now, the question is asking whether the little girl has very thin legs. Looking at the image, her legs do appear to be on the slender side, but I need to be careful here. Children naturally have smaller body proportions compared to adults, and it's normal for their limbs to look thin, especially if they're active and healthy. I should also consider the perspective of the photo. Since she's standing and the photo is taken from a distance, her legs might appear thinner due to the angle and the way the light is hitting them. Additionally, the clothing she's wearing— black leggings and boots—might make her legs look thinner than they actually are because of the color and fit of the clothes. Another thing to consider is the resolution of the image. If the image isn't very high resolution, details might be lost, making her legs appear thinner than they are in reality. But based on what I can see, her legs do appear to be quite thin. However, without knowing her age, height, weight, or overall health status, it's difficult to make a definitive judgment about whether her legs are "very thin " in a concerning way. Perhaps it's better to approach this from a descriptive angle rather than making a value judgment. So, objectively, her legs are slender or thin, but whether that's "very thin " is subjective and depends on the context and the viewer's perspective. Alternatively, maybe the question is testing my ability to observe details accurately and make an assessment based on visual information alone. In that case, I would say that yes, her legs appear to be very thin based on the image provided. Wait a minute, maybe I should think about this differently. If I were to describe her appearance to someone else, I would mention that she has very thin legs. But if I were to assess her health based on this image, I wouldn't have enough information to make that kind of determination. I should also consider if there's any possibility of distortion in the photo. For example, if the camera was zoomed in too much or if there was a wide-angle lens used, that could affect the appearance of her legs. Looking back at the image, I can see that the photo seems to be taken from a normal distance, without any obvious distortion. The proportions of her body relative to her surroundings seem consistent. Another angle to consider is cultural perceptions of body image. What might be considered "very thin " in one culture might be seen as healthy or normal in another. But since the question is presented in an English language test, presumably in a Western context, I should base my answer on general perceptions in such cultures. Moreover, in medical terms, being "very thin " could imply underweightness, which might be a health concern. However, again, without knowing her actual measurements, it's impossible to make such a determination from an image alone. Perhaps the question is simply testing the ability to observe and describe what is visible, without overanalyzing or making assumptions beyond what the image presents. In that case, my answer would be that yes, the little girl has very thin legs, as observed in the image. But to be thorough, I should consider if there's any possibility that the image could be misleading. For instance, if she's standing on tiptoes, her legs might look thinner, but from the photo, it seems like she's standing flat-footed. Also, the fit of her clothing is important. If her leggings are too big for her, they might make her legs look thinner than they are. However, from the image, the leggings seem to fit snugly around her legs, so that's probably not the case. Another consideration is the lighting. If the lighting is harsh or if there are shadows cast in a certain way, that could affect the appearance of her legs. But the photo is taken outdoors on a sunny day, so the lighting should be natural and even, minimizing any distorting effects. Furthermore, the resolution of the image appears to be adequate for making out details, so I don't think that's an issue here. So, after considering all these factors, I would conclude that yes, the little girl in the image has very thin legs. **Final Answer: A) Yes**

**Perception error:** Although some models realize that the girl's legs are unusually thin and suggest that this phenomenon "may not be in line with common sense", most of them still tend to attribute it to the visual compression effect caused by dark or tight clothing, or conduct semantic analysis on the concept of "thinness" itself, and fail to really find the core factor that causes the illusion - the white popcorn bag in the girl's hand. This bag blocks the area between the legs, causing edge visual fusion, thus creating the illusion of unusually thin legs. However, the model does not examine the occluded area through local magnification or structural analysis during the reasoning process, and only judges based on surface features. Although models such as QVQ or Gemini-flash-thinking attempt to carry out detailed conceptual analysis, under the premise of limited visual perception quality, this reasoning path based on common sense and language is prone to mislead judgment. For the One-Pass multimodal reasoning model, once the initial visual perception result is biased, the subsequent text reasoning will be carried out along the wrong path, lacking a correction mechanism, and ultimately leading to wrong conclusions.

Figure 7: A sample error case of BLINK-Twice (Do the little girl in the image have very thin legs?).

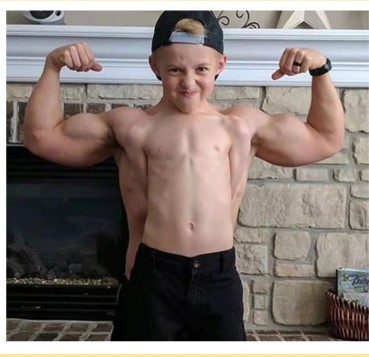

### ❓ Question

**Does the child in the image have large muscles?**

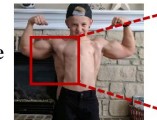

### 🔍 Detailed Visual Clues

While the muscular arms appear to belong to the child, a closer look reveals that his real arms are hidden behind his back, with parts of them still faintly visible. The large arms are aligned from behind, and do not naturally connect to the child's shoulders.

### ✅ True Reality

An adult is standing behind the child and flexing their arms in perfect alignment with the child's body. The child's own arms are tucked behind, creating a visual illusion that the large muscles belong to him.

---

**GPT-4o**: To determine whether the child in the image has large muscles, follow these steps: 1. **Observe the Muscles**: Look at the size and definition of the arms, chest, and overall physique. 2. **Compare to Typical Child Development**: Consider the typical muscle size for children in this age range. Usually, children do not have highly developed muscles unless they are engaged in specific strength training or bodybuilding activities. 3. **Assess Muscle Size**: The child in the image shows significantly developed muscles, especially in the arms, which are larger and more defined than one would typically expect for a child. 4. **Final Assessment**: Based on the observation that the muscles are visibly large and well-defined, it can be concluded that the child does have large muscles. **Final Answer: A) Yes**

---

**Qwen2-VL-72B**: To determine if the child in the image has large muscles, let's analyze the visual evidence step by step: 1. **Posture and Flexing**: The child is flexing both arms, which is a common pose to display muscle size. This suggests an intention to show off muscle mass. 2. **Muscle Definition**: The biceps and triceps are visibly defined and appear larger than what would be typical for a child of similar age. The definition and size suggest that these muscles are well-developed. 3. **Body Proportions**: The child's upper body, particularly the arms, appears disproportionately larger compared to the rest of the body. This further supports the idea that the muscles are larger than average. 4. **Contextual Clues**: The child is shirtless, which is often done to showcase physical attributes like muscle mass. The presence of a watch and a cap does not detract from the visible muscle size. Given these observations, it is reasonable to conclude that the child does have large muscles. **Final Answer: A) Yes**

---

**Gemni-2.0-flash-thinking**: **Rationale:** The image depicts a young boy flexing his arm muscles. Looking at his biceps, triceps, and shoulder muscles, they appear to be well-defined and larger than what is typically expected for a child of his apparent age. The muscle definition is clearly visible, indicating a significant level of muscle development. While it's important to consider natural variation, in this image, the child's muscles are noticeably prominent. Answer: A) Yes **Final Answer: A**

**QVQ**: So I've got this image of a young boy who's flexing his muscles. He's shirtless, wearing a cap backwards, and has a watch on his left wrist. He's making a face that suggests he's straining or showing off his strength. The background shows a fireplace with a mantel that has some decorative items, including a star and what looks like a coffee cup. Now, the question is asking whether this child has large muscles. The options are "Yes" or "No." I need to determine which one is correct. First, I should consider what constitutes "large muscles" for a child. Muscular development varies based on age, activity level, and individual differences. Since he's flexing and appears to have well-defined biceps and triceps, it might be tempting to say "Yes." However, I should also consider if this might be a normal variation or if there could be other factors at play. For example, some children might have more muscle tone due to sports or physical activities, while others might not. Another thing to consider is the possibility of digital manipulation. In today's world, images can be easily edited to enhance or exaggerate certain features. The proportions of the muscles seem somewhat disproportionate to the rest of his body, which might suggest that the image has been altered. Additionally, I should think about the ethical implications of accepting or rejecting the idea that a child has large muscles. It's important to approach this with sensitivity and avoid making assumptions based on appearance alone. Perhaps I should look for more context or additional information about the image. Maybe there's a story behind it that could provide clarity. For instance, maybe the child is involved in sports or fitness activities that have led to increased muscle development. On the other hand, it's also possible that the image is intended to be humorous or provocative, and the muscles are exaggerated for effect. Considering all this, I need to make an informed decision based on what I can see in the image. Let me try to analyze the muscles more closely. His biceps are particularly prominent, and his triceps are also well-defined. His shoulders and chest also appear to have more muscle mass than typical for a child his age. However, without knowing the baseline for muscle size in children of his age and build, it's challenging to make an accurate judgment. Maybe I should consider comparing him to other children of similar age. But since I don't have any reference points in the image, that's not possible. Another angle to consider is the possibility of a medical condition that causes increased muscle mass. There are rare conditions where children have abnormally large muscles, but without medical knowledge or context, it's not appropriate to speculate on that. Furthermore, I should consider the child's overall health and well-being. Having large muscles doesn't necessarily indicate good health, and it's not my place to make assumptions about his health based solely on this image. Perhaps the best approach is to focus solely on the visual evidence presented in the image and make a straightforward assessment. In the image, the child is clearly flexing his muscles, and they appear to be larger and more defined than what might be expected for his age. Therefore, based on the visual information provided, it seems reasonable to answer "Yes," the child has large muscles. However, I must acknowledge that this conclusion is based solely on the appearance in the image and without any verifiable context or additional information. Alternatively, if I consider that the image might be digitally altered or that the appearance is not representative of the child's actual muscle size, I might lean towards answering "No." But without evidence to support either scenario, it's difficult to be certain. In such a case, perhaps the most objective approach is to go with the apparent visual reality presented in the image. Thus, my final answer would be "Yes." **Final Answer:** A) Yes

**Perception error:** The model mistakenly identified the exaggerated muscular arms as the boy's own limbs. The fundamental reason is that it did not recognize the lack of physical connection between the muscular arms and the boy's shoulders, ignored the details of the real arms hidden but visible behind him, and the obvious inconsistency between the boy's body proportions and muscle volume. The model took a path based on common sense and biological inference (i.e. "Children usually don't have muscles → but this child seems to have → so he may have trained"), rather than judging based on the image structure itself.

Figure 8: A sample error case of BLINK-Twice (Does the child int the image have large muscles ?).

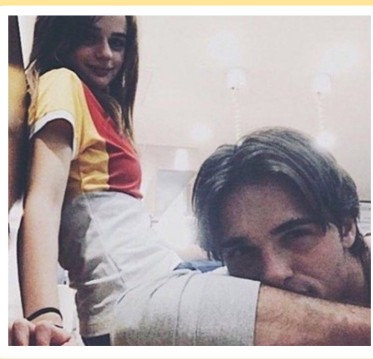

### ❓ Question

**Is the man's head resting on the woman's legs?**

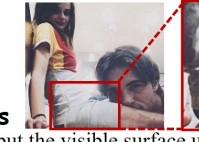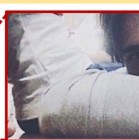

### 🔍 Detailed Visual Clues

The girl is wearing blue jeans, but the visible surface under the man's head is a gray, wrinkled fabric, which belongs to the man's sleeve. The texture and fold lines on the sleeve clearly indicate it is not denim. Additionally, the man's forearm is visible, showing that his head is resting on his own arm.

### 📄 True Reality

The camera angle and overlapping positions make it seem like the man's head is resting on the girl's legs, when in fact it is resting on his own arm.

---

**GPT-4o**: To determine if the man's head is resting on the woman's legs, let's analyze the image step by step: 1. **Identify the Elements in the Image**: - There is a woman sitting, likely on a surface like a table or counter. - A man is positioned with his head near the woman's legs. 2. **Observe the Positioning**: - The man's head appears to be leaning towards or against the woman's legs. - The positioning suggests proximity and contact between the man's head and the woman's legs. 3. **Evaluate the Contact**: - The alignment of the man's head and the woman's legs seems to indicate that his head is indeed resting on her legs. - There is no visible gap between the man's head and the woman's legs, implying contact. 4. **Consider the Context**: - The expression and body language of both individuals do not suggest tension, supporting the idea of a resting position. Based on the analysis, the man's head appears to be resting on the woman's legs. **Final Answer: A) Yes**

---

**Qwen2-VL-72B**: To determine if the man's head is resting on the woman's legs, let's analyze the image step by step: 1. **Identify the positions of the man and the woman**: - The woman is sitting upright. - The man is lying down. 2. **Examine the position of the man's head**: - The man's head is positioned near the woman's lap. 3. **Check if there is any physical contact between the man's head and the woman's leg**s: - The man's head appears to be resting on or very close to the woman's legs. Based on this analysis, it seems that the man's head is indeed resting on the woman's legs. **Final Answer: A) Yes**

---

**Gemni-2.0-flash-thinking**: **Rationale: Rationale**: By carefully observing the original image and the provided crops, we can analyze the relative positions of the man's head and the woman's legs. * **Positioning**: The man's head is located directly adjacent to and slightly below the woman's legs. His head appears to be in contact with the leg area.* **Perspective**: While the image is taken at an angle, it's clear that the man's head is not just near, but physically against the woman's legs. There is no visible gap to suggest his head is floating or positioned elsewhere.* **Body Language**: The pose suggests a relaxed and close interaction, which is consistent with someone resting their head on another person's legs. Considering these observations, it is highly likely that the man's head is indeed resting on the woman's legs. **Final Answer: A) Yes**

**QVQ**: Alright, I've got this image to describe. Let's see, there are two people in it, a man and a woman. The woman is standing, and she's wearing a yellow and red striped shirt with white shorts. She has long hair and is smiling at the camera. The man is sitting down, resting his head on what appears to be the woman's legs. He's wearing a gray long-sleeve shirt and has dark hair. The background is a bit blurred, but it looks like an indoor setting with some lights hanging from the ceiling. So, the main thing here is the position of the man's head. It seems like he's lying down or sitting in such a way that his head is supported by the woman's legs. The question is asking specifically whether his head is resting on her legs. Let me think about this. If he's sitting, and she's standing, and his head is on her legs, that would make sense. Maybe she's holding his head in her lap or something similar. Alternatively, perhaps he's lying down on a couch or something, and she's sitting next to him with her legs crossed, and his head is on her knee. But from the description, it says she's standing, so maybe he's sitting on the floor, and she's standing over him with her legs crossed, and his head is on her knees. That could be a possible arrangement. Another possibility is that he's lying down on a bed or a couch, and she's standing next to him, and he's resting his head on her legs. But that seems a bit awkward for her to be standing like that for an extended period. Alternatively, maybe she's sitting on a chair or something, and he's sitting on the floor with his head on her legs. That seems plausible. Wait, but the description says she's standing. So, if she's standing, and he's sitting or lying down with his head on her legs, that would make sense. Let me consider the context. They seem to be in a cozy, indoor setting, maybe at home or in a hotel room. The lights in the background suggest a relaxed atmosphere. Looking back at the question: "Is the man's head resting on the woman's legs?" Given the positions described, it seems likely that his head is indeed resting on her legs. The alternative would be that his head is on a pillow or something else, but the description doesn't mention any pillows. Perhaps they are relaxing together, and he's resting his head on her legs for comfort. It's a intimate gesture, suggesting they have a close relationship. I should also consider the expressions on their faces. The woman is smiling at the camera, while the man appears to be looking down, possibly at something in his hands or just resting. His expression seems relaxed. So, based on the description, it appears that the answer is yes, the man's head is resting on the woman's legs. Wait a minute, the answer options are: A) Yes B) No Given that the description says the man is resting his head on what appears to be the woman's legs, and considering the positions they are in, it's reasonable to conclude that the answer is A) Yes. However, I should consider if there's any possibility that his head isn't on her legs. Maybe he's resting his head on a pillow or on the couch, and her legs are just positioned nearby. But the description specifically mentions that his head is on what appears to be her legs, so it's likely that it's directly on her legs. Another thought: perhaps he's lying down, and she's standing next to him with her legs crossed, and his head is on her knees. That seems like a plausible arrangement. Alternatively, if she's sitting, and he's sitting on the floor with his head on her legs, that also fits the description. But again, the description says she's standing, so the first scenario is more likely. In any case, based on the given information, it's reasonable to conclude that the man's head is resting on the woman's legs. So, my final answer is A) Yes. **Final Answer: A) Yes**

**Perception error:** Although these models all try to judge the content of the image through step-by-step reasoning, they fail to recognize that the elbow below the man's head is not the woman's thigh due to the limitation of the one-pass perception mechanism, and thus make a wrong judgment. The model tends to focus on the posture of the characters and the contact position of the surface, but ignores key details, such as the wrinkles of the gray sleeves and the blue jeans worn by the woman, which are clues that can be easily identified through local magnification. They rely more on intuitive analysis of "looks like" in the reasoning process, without the necessary local review or detail confirmation, resulting in misunderstanding of the image content. This problem highlights the lack of correction mechanism of the current model under the initial perception bias, and in the future, it should consider introducing multiple perceptions or local magnification to improve the accuracy of judgment.

Figure 9: A sample error case of BLINK-Twice (Is the man's head resting on the woman's legs?).