# OpenReview forum: "BLINK-Twice: You see, but do you observe?  A Reasoning Benchmark on Visual Perception"
_NeurIPS.cc/2025/Datasets_and_Benchmarks_Track — NeurIPS 2025 Datasets and Benchmarks Track poster_

### Official Review · Reviewer_iirm · 2025-06-17

**Rating:** 5
**Confidence:** 4

**Summary:**

This work focuses on evaluating the reasoning abilities of multimodal large language models (MLLMs) with an emphasis on visual understanding. Most existing benchmarks assess reasoning primarily through language, often treating the visual input as optional or interchangeable. To address this gap, the authors introduce BLINK-Twice, a dataset designed to challenge models with seven types of visually grounded reasoning tasks. It contains naturally occurring adversarial image pairs that require genuine visual comprehension and includes annotated reasoning chains to assess not just the final answer, but the reasoning process itself. Experiments reveal that while language-based reasoning techniques, such as chain-of-thought prompting or self-criticism, can boost performance, they often lead to unstable or overly verbose outputs. In contrast, encouraging models to revisit the image improves reasoning consistency and overall performance.

**Additional Feedback:**

- The discussion on the limitation (difficulty-level distinctions, multilingual MLLMs, and video) is good. It would be better to include it in the conclusion section.

**Dataset Code Accessibility:**

Yes

**Dataset Code Comments:**

I have checked the code, which is mostly clear, and the script is provided to evaluate other MLLMs

**Ethical Considerations:**

No, there are no or only very minor ethics concerns

**Final Justification:**

Thanks for the response, which addressed all my initial concerns. Please incorporate it into the revised version.

**Limitations Weaknesses:**

- [***Section 4.3 (Reasoning Model Analysis) needs clarification***] Section 4.3 Reasoning Model Analysis is not very clear. 1) Figure 4(a) needs clarification: add labels for the reasoning models. 2) The visualization of the reasoning process is hard to understand. What is the main point of Figure 4(b)? Also, the self-criticism is not well introduced.
- [***BLINK vs. BLINK-Twice***] The introduction briefly illustrates the difference from BLINK (Lines 49–50). However, it is still not clear how BLINK-Twice differs from BLINK. The latter contains multiple tasks that share some commonality with tasks in BLINK-Twice. Please provide a detailed clarification on the connection and highlight the differences.
- [***Discuss Seven reasoning types***] The seven types are good, but what are the statistics of each type, such as how many images are in each type and the performance of MLLMs on each? Is there any correlation between these types, such that if MLLMs perform poorly on one type, they also perform poorly on another? This may provide more insights.
- [***Beyond Section 4.4 (multi-turn)***] Section 4.4 shows that multi-turn could help, which is reasonable. Is there any other ways to improve the performance? Please share some suggestions or comments.

**Strengths Contributions:**

+ The motivation is sound. It is good to evaluate the reasoning capability of MLLMs on visual perception. Seven types of tasks are interesting: they force the MLLMs to rely on visual input for reasoning.

+ Adversarial sample generation is insightful. The pair of data is useful and nicely demonstrated.

+ The experiments are clear and well-presented. Most of MLLMs are covered, and this submission provides evaluation on chain-of-thought prompting and self-criticism. Also, revisiting the image is shown to be useful.

---

> ### Author Rebuttal · Authors · 2025-07-30
>
> **Q1.** We appreciate the reviewer’s feedback regarding Section 4.3 and Figures 4(a) and 4(b). Due to rebuttal constraints, we are unable to include the revised figures directly, but we provide a detailed textual clarification below:
>
> - (1) Figure 4(a): In the original figure, each pair of bars represented a base MLLM (left) and its corresponding reasoning-augmented version (right)—e.g., Qwen2-VL 72B vs. QVQ-72B, Claude 3.7 vs. Claude 3.7-thinking. We fully agree that this distinction was not visually clear. In the revised figure, we add striped shading to all reasoning-augmented models and include an explicit legend to clearly distinguish between base and reasoning-enhanced models.
>
> - (2) Figure 4(b): The initial version attempted to show reasoning traces from both Qwen and Gemini, which diluted the main message due to space constraints. In the revision, we now focus solely on Qwen to maintain clarity. The example highlights self-criticism behavior—e.g., a mid-chain statement such as “Wait a minute, maybe I should think about this differently...”, indicating the model’s capacity for internal revision. This moment is explicitly marked to illustrate the introspective reasoning that reasoning-enhanced models can exhibit.
>
> We believe these revisions improve the interpretability and communicative clarity of our reasoning model analysis, and we will include them in the final version.
>
> ---
>
> **Q2.**  We thank the reviewer for the thoughtful question. While BLINK and BLINK-Twice share the goal of evaluating MLLMs’ visual capabilities, they differ fundamentally in the level of reasoning required. As explicitly stated in the BLINK paper, “most of the BLINK tasks can be solved by humans within a blink.” These tasks often involve direct visual perception, such as counting (e.g., “How many people are preparing food?” , as shown in our Fig. 1(a).) or VQA questions that rely on explicit visual evidence. BLINK also includes some simple spatial reasoning, such as “Is the bed at the right side of the dining table?”—questions that can be answered with a quick scan of the image, without requiring deeper visual inference.
>
> In contrast, BLINK-Twice is designed to go beyond what can be perceived ‘within a blink.’ The tasks demand attentive visual scrutiny and reasoning over subtle cues, making them cognitively more challenging. For instance, in the question “Is there a cat and its shadow?”, a quick impression may lead to a wrong answer (“yes”), but a closer observation reveals that the so-called “shadow” actually has ears and eyes—indicating it's a second black cat. Correctly answering such questions requires careful observation and inference, rather than superficial perception.
>
> Therefore, while both BLINK and BLINK-Twice evaluate the visual capabilities of MLLMs, BLINK primarily focuses on perceptual understanding, whereas BLINK-Twice requires models to jointly perform visual perception and reasoning.
>
> ---
>
> **Q3.** We thank the reviewer for the thoughtful suggestion. In the current version, we only presented the distribution of the seven reasoning types in Fig. 2(a) via the outer ring of the donut chart, without providing explicit image counts or model-wise performance breakdowns. In the revised version, we will include a table summarizing the number of samples per category as well as representative model performance on each type. Overall, we observe that MLLMs exhibit relatively balanced performance across most reasoning types, without any category showing extreme highs or lows.
>
>
> | Category              | Number | QVQ-72B-preview | Qwen2.5-VL-32B | Gemini-2.0-flash | GPT-4o | O1    |
> |-----------------------|--------|------------------|------------------|-------------------|--------|--------|
> | Visual Misleading     | 58     | 0.5809           | 0.6569           | 0.7313            | 0.6715 | 0.7122 |
> | Forced Perspective    | 30     | 0.5615           | 0.5714           | 0.5865            | 0.5564 | 0.6288 |
> | Physical Illusion     | 20     | 0.5938           | 0.6471           | 0.6571            | 0.6176 | 0.6286 |
> | Visual Dislocation    | 130    | 0.5886           | 0.5966           | 0.5657            | 0.5852 | 0.6171 |
> | Art Illusion          | 35     | 0.5233           | 0.6322           | 0.5349            | 0.7126 | 0.5795 |
> | Visual Occlusion      | 52     | 0.5993           | 0.6105           | 0.5560            | 0.5880 | 0.6232 |
> | Motion Illusion       | 20     | 0.6364           | 0.7273           | 0.5652            | 0.7727 | 0.7391 |
>
>
>
> To further explore inter-type relationships, we computed Pearson correlation coefficients between the category-wise scores across all models. Notably, Visual Dislocation and Visual Occlusion show relatively high correlation (0.853), likely because both involve reasoning over overlap, occlusion, or contour fusion. In contrast, Motion Illusion shows low correlation with other types (the bottom row), which may be attributed to its distinct underlying causes compared to the rest. We will include the full quantitative results and correlation analysis in the supplementary material of the final version to provide deeper insight into model behavior across reasoning types.
>
>
> |                        | Visual Misleading | Forced Perspective | Physical Illusion | Visual Dislocation | Art Illusion | Visual Occlusion | Motion Illusion |
> |------------------------|-------------------|---------------------|--------------------|---------------------|---------------|-------------------|------------------|
> | Visual Misleading      | 1.000             |                     |                    |                     |               |                   |                  |
> | Forced Perspective     | 0.780             | 1.000               |                    |                     |               |                   |                  |
> | Physical Illusion      | 0.786             | 0.691               | 1.000              |                     |               |                   |                  |
> | Visual Dislocation     | 0.795             | 0.788               | 0.606              | 1.000               |               |                   |                  |
> | Art Illusion           | 0.690             | 0.527               | 0.702              | 0.667               | 1.000         |                   |                  |
> | Visual Occlusion       | 0.743             | 0.760               | 0.796              | **0.853**               | 0.707         | 1.000             |                  |
> | Motion Illusion        | 0.611             | 0.503               | 0.632              | 0.574               | 0.652         | 0.708             | 1.000       |
>
> ---
>
>
> **Q4.**  We thank the reviewer for the helpful suggestion. Beyond multi-turn inputs, another promising direction is enhancing MLLMs with external visual tools, such as depth estimators, segmentation models, and grounding modules, which provide structured cues to improve visual understanding. For example, Visual Sketchpad equips models with a sketching interface to support visual chain-of-thought reasoning.
> In addition, training-time interventions can further boost reasoning ability. Recent methods like Pixel Reasoner encourage models to perform active perceptual operations (e.g., zooming, focusing) via reinforcement learning, enabling better spatial reasoning.
>
> ---
>
> **Q5.**  We thank the reviewer for the helpful suggestion regarding the presentation of our limitation discussion. While we currently include detailed analysis of limitations—such as difficulty-level distinctions, multilingual MLLMs, and video-based reasoning—in a dedicated section of the supplementary material, we agree that summarizing these points in the conclusion would improve the clarity and completeness of the main paper. We will revise the conclusion accordingly in the final version.

---

> > ### Comment · Reviewer_iirm · 2025-08-05
> >
> > Thank you for the response, which addressed all my concerns. I therefore maintain my original rating of Accept.
> > In the meantime, I have reviewed the discussion with Reviewer 8pg1 and found it constructive. Please consider incorporating the key clarifications and points from that exchange into the revision as well.

---

> > > ### Author Response · Authors · 2025-08-05
> > >
> > > Dear Reviewer iirm,
> > >
> > > Thank you very much for your thoughtful review and for taking the time to carefully engage with our work. We truly appreciate your positive feedback and are glad to hear that your concerns have been fully addressed.
> > >
> > > As you suggested, we will incorporate the key clarifications and insights from the discussion with Reviewer 8pg1 into the final version of the paper to further improve its clarity and impact.
> > >
> > > Thank you again for your time and valuable input.
> > >
> > > Sincerely,
> > > The Authors

---

### Official Review · Reviewer_8pg1 · 2025-06-18

**Rating:** 4
**Confidence:** 5

**Summary:**

The paper proposes a new benchmark, BLINK-Twice, which contains 345 images and 650 VQA samples spanning 7 types of visual challenges that require visual reasoning. Using this benchmark, the authors ablate over 20 high-performing multimodal language models, including commercial ones such as Gemini-2.0 and Claude, to analyze their failure modes. The paper concludes with interesting findings about the visual reasoning behaviors of current models and suggests promising directions for improvement.

**Additional Feedback:**

I encourage the authors to improve the Hugging Face dataset viewer so that reviewers and users can more easily assess the quality and diversity of the benchmark. Overall, I find the benchmark interesting and potentially useful, but several aspects—such as overclaiming and missing evaluation and curation details—need to be clarified. I would be very open to reconsidering my score if the authors are able to address these concerns.

**Dataset Code Accessibility:**

Yes

**Dataset Code Comments:**

The huggingface repo provides the full dataset and evaluation code.

**Ethical Considerations:**

No, there are no or only very minor ethics concerns

**Final Justification:**

The authors have addressed most of my concerns regarding the lack of evaluation and licensing details, though I still have reservations about the small dataset size and overclaiming. I lean toward acceptance, given their willingness to engage with feedback. Since many statements, numbers, and figures in the dataset require updates, I encourage a thorough revision to ensure the paper is polished, regardless of the decision. The claims should also be toned down, as the current evidence does not justify that better reasoning necessarily improves benchmark performance. Due to this year’s NeurIPS policy, the dataset cannot be updated during review, so my judgment and rating is based on the current state.

**Limitations Weaknesses:**

I have some questions regarding the benchmark curation and evaluation protocols:

1. The calculation of Q-Acc appears to differ from that used in NaturalBench. In line 195, it states that Q-Acc is defined as “either question correct,” whereas the original Q-Acc requires both images per question to be answered correctly.

2. The paper observes that presenting the same image multiple times can enhance performance and suggests that active perception (e.g., zooming in) could improve results further. However, the logical connection between these two findings is unclear. Moreover, the reasoning traces in this dataset appear to be purely textual—there is no visual zoom-in or rotation involved.

3. I have some concerns about the size of the dataset (only <400 images).

4. The human verification process is not clearly explained. The paper mentions using GPT-4o to generate reasoning traces for each image, which are then verified by humans. But it is unclear whether humans correct any errors or redundancies in the traces, or if they simply discard flawed examples.

5. How was GPT-4o used, exactly? As far as I know, the API version does not support visual zoom-in or other forms of active perception, which differs from the web version. This distinction should be clarified in the main paper.

6. Are the skill tags overlapping? For instance, visual dislocation, occlusion, and forced perspective appear very similar. Examples such as:
– A man standing in front of a tree appears to have an exaggerated “afro” due to the foliage alignment.
– A dog blocking a man's face creates the illusion of a “dog head mask.”
– A golf ball close to the lens appears enormous while a distant man appears to hold it.
All seem to involve visual spatial reasoning, raising questions about the distinctiveness of the skill categories.

7. The authors should have disclosed the use of GPT-4o for generating reasoning traces in the NeurIPS checklist, specifically in response to Question 16.

8. Line 64: The statement “Current MLLMs exhibit potential flaws in visual reasoning, often ‘see’ but fail to ‘observe’” is vague. How do you determine whether a model is “seeing” but not “observing”? This claim needs further justification.

9. Supplement Lines 9–17: It is unclear how the authors filtered from 15,000 to only 600 images. Please provide the exact prompts used. It is also confusing how the authors claim to have selected only natural scenes, yet some dataset images clearly depict abstract or stylized content (e.g., comic-style instruction manuals). The image selection process needs clarification.

10. The license information for the images is not mentioned in either the main paper or the supplement. This is important, as some sources (e.g., Shutterstock/Getty Images) prohibit the use of their content in AI training or evaluation (if you don't purchase their images). The paper should clarify how it handles licensing issues to ensure the benchmark can be legally and broadly adopted by the research community.

11. I highly recommend improving the Hugging Face dataset viewer. Currently, it only displays the images. Including the corresponding questions and answers would greatly help others in verifying the benchmark’s quality.

12. Rather than positioning this as a "visual reasoning" benchmark, it may be more accurate to describe it as a visual perception puzzle dataset. Many of the examples are visually challenging—even for humans. For example, I found myself initially misled by the image with foliage forming an exaggerated “afro.” A more formal definition of “visual reasoning” would be helpful. For instance, NaturalBench categorizes skills into object recognition, attributes, relationships, and higher-order reasoning (e.g., negation and universal). In contrast, solving BLINK-Twice often feels more like handling difficult perception tasks than reasoning per se.

**Strengths Contributions:**

The benchmark appears natural and engaging, with sufficient challenges for leading models (suggested by the low overall performance). The model ablations and qualitative examples provided in the paper are reasonable and help readers better understand the nature of the benchmark. The paper is well-written, and most of the data collection details are clearly disclosed in the paper or the supplemental. All data and questions are openly available on the Hugging Face website. That said, I would recommend that the authors improve the benchmark visualization to make it more compelling and accessible to the broader research community.

---

> ### Author Rebuttal · Authors · 2025-07-30
>
> **Q1.** We sincerely thank the reviewer for their careful attention to evaluation details.  We acknowledge that our previous computation of Q-Acc differs from the definition in Natural-Bench, which requires both images in each question to be correctly answered. Our original Q-Acc is closer to Overall Accuracy, reflecting the average correctness across all questions. We will rectify this in future versions by reporting the Q-Acc consistent with Natural-Bench and renaming the current metric as Overall Acc. Due to space limitations, we present below the Q-Acc results (as defined in Natural-Bench) for selected models only:
>
>
> |  | InternVL2-40B | Qwen2.5-vl-72b | Claude 3.7-thinking | GPT-4o | Gemini-2.5-pro |
> | ---     | ---           | ---            | ---                 | ---    | ---            |
> | Q-Acc   | 0.3529        | 0.3861         | 0.3594              | 0.4059 | 0.5111         |
>
> ---
>
> **Q2.**  We appreciate the reviewer’s observation and the opportunity to clarify two points: the connection between repeated viewing and active perception, and that the reasoning traces do not involve explicit visual operations.
>
> ***(1) Repeated Viewing and Active Perception.*** Both strategies fall within the broader scope of multimodal visual reasoning, as detailed in Supplementary Section B.3 (Lines 107–124, Fig. 6), where we outline a three-level framework for perception-based reasoning:
> - Level 1: One-Pass Perception– the image is processed once during encoding, and all subsequent reasoning is purely language-driven, as in most current MLLMs.
> - Level 2: Re-Perception – the model is guided to revisit the image in a multi-turn setup based on textual feedback, as demonstrated in our **repeated viewing** experiment.
> - Level 3: Collaborative Perception – the model itself initiates visual perceptual actions (e.g., zooming, rotating) to support inference, aligning with the notion of **active perception** discussed in our paper.
>
> In our experiments, we employed multi-turn interactions to enforce repeated viewing, prompting the model to revisit the visual input. This led to improved reasoning performance compared to Level 1. Based on this observation, even passive repetition (Level 2) enhances reasoning. In contrast, active perception (Level 3) enables not only multiple rounds of visual access but also allows the model to autonomously perform targeted perceptual actions (e.g., zooming, rotating), leading to finer-grained understanding and more effective evidence gathering. We believe this could further improve model performance. We will clarify the connection between these two strategies more explicitly in future versions.
>
>
> ***(2) Textual Reasoning paths.*** In our dataset, the reasoning paths do not include explicit visual operations, because our goal is to assess visual reasoning capability, rather than the model’s ability to actively perform visual manipulations.As shown in Fig. 1 (c), although the reasoning does not mention “zooming into the wheel-ground contact area,” it does include fine-grained visual cues, such as lighting conditions and contact evidence. BLINK-Twice allow models to reason correctly by directly observing the image, without requiring explicit visual actions.This design ensures the evaluation focuses on the model’s capacity to extract and reason over visual information, not its interactive interface or manipulation ability.
>
> That said, we recognize the value of incorporating visual operations. In future work, we plan to annotate each image with suggested perceptual steps—e.g., “zoom into region X, then rotate”—to support the development and evaluation of models with interactive visual reasoning capabilities.
>
> ---
>
> **Q3.** We thank the reviewer for highlighting the importance of dataset size. While our dataset is relatively small, it is carefully curated to span diverse and challenging visual reasoning scenarios. The size limitation arises from the high cost and difficulty of collecting and annotating such complex images.  To further enhance the reliability of our evaluation, we report variance estimates via repeated sampling (see Reviewer e2J2). We recognize the value of scaling up and plan to expand the dataset in future work.
>
> ---
>
> **Q4.** We appreciate the reviewer’s attention to the human verification process. For each GPT-4o-generated reasoning trace that was factually incorrect or logically flawed, human annotators were instructed to revise and correct the trace rather than simply discard it. We will clarify this process in the revised paper.
>
> ---
>
> **Q5.** Indeed, GPT-4o does not support active visual operations such as zoom-in. As noted in Lines 280–283 and in the caption of Fig. 6, the visual interaction results were obtained from the o3 web interface, not from GPT-4o. We will make this distinction more explicit in the revised version to avoid confusion.
>
> ---
> **Q6.** We thank the reviewer for the insightful comment. We agree that visual dislocation and visual occlusion share perceptual similarities, such as spatial overlap and contour fusion, and that Fig. 2 may not have clearly distinguished them. To clarify: visual dislocation arises from spatial alignment that creates illusory structures (e.g., foliage forming an “afro”), while visual occlusion involves obstruction that hides parts of an object (e.g., a pillar concealing an arm, creating an “amputated” appearance). Forced perspective is more conceptually distinct, relying on camera angles and depth cues to alter perceived size or position (e.g., an enlarged seagull due to lens proximity). We acknowledge that some cases—like the “dog head mask”—could reasonably fit under either dislocation or occlusion. In future work, we may consider merging these categories.
>
> ---
>
> **Q7.** We appreciate the reviewer’s suggestion and agree on the importance of transparency. We will revise the checklist to explicitly disclose our use of GPT-4o for generating reasoning traces and conducting CoT-Score evaluations in response to Question 16.
>
> ---
>
> **Q8.** We thank the reviewer for pointing this out. In our paper, we distinguish between “seeing”—referring to surface-level perception and “observing”, which requires attending to subtle visual cues and reasoning over them in context. BLINK-Twice is specifically designed to evaluate the latter. As shown in our results, current MLLMs, including GPT-4o, perform poorly on such tasks (e.g., I-Acc < 0.4 and G-Acc < 0.2), indicating limitations in fine-grained visual reasoning. We will clarify this in the revised version.
>
> ---
>
> **Q9.** Thank you for your valuable feedback. To clarify, we used GPT-4o to filter out classic optical illusions *（Supplementary Fig 1 (a)）* such as the Müller-Lyer illusion (arrows on the ends of lines creating length misperceptions) and the Ebbinghaus illusion (a central circle appearing larger or smaller depending on surrounding circles), etc. Our filtering guideline overview was as follows:
>
>
> > You are an expert in cognitive science and visual perception. Your task is to determine whether a given image description involves any classic optical illusions, rather than naturally captured images. These include well-known illusions such as the Ebbinghaus illusion, Müller-Lyer illusio....
> >  Output format: Classic illusion: Yes/No
>
>
> These illusions primarily test whether the model is misled by perceptual biases (e.g., psychological illusions), rather than engaging in genuine visual reasoning.  That said, a small number of images with abstract or stylized appearances were retained, as they do not fall under classic optical illusions. We appreciate the reviewer’s suggestion and will revise our description to more clearly articulate this distinction and Supp-Fig 1.
>
> ---
>
>
> **Q10.** We thank the reviewer for raising this important issue. We acknowledge that due to an oversight during the initial data collection phase, a small number of images (<10) from Getty Images were mistakenly included in the dataset. These have been removed and will be replaced with properly licensed alternatives, along with updated evaluation results. We sincerely apologize for this oversight. To ensure legal compliance and transparency, the final dataset will include the source URL and licensing documentation for each image to prevent future issues.
>
> ---
>
>
> **Q11.** We sincerely thank the reviewer for this valuable suggestion. We fully agree that displaying questions and answers alongside the images in the Hugging Face dataset viewer would significantly improve transparency and usability. Due to the constraints of the rebuttal phase, we are unable to update the repository immediately, but we will make this enhancement in the final version.
>
>
> ---
> **Q12.** We sincerely appreciate the reviewer’s insightful and constructive feedback. As you rightly pointed out, many examples in BLINK-Twice are indeed visually challenging—even within our own team, certain samples have sparked intense discussions.
>
> We agree that viewing the dataset as a visual perception puzzle dataset is a reasonable and valid perspective. However, we define it as a visual reasoning dataset based on the capabilities it aims to evaluate, rather than the surface form of the questions. While the task format may resemble a visual perception puzzle, it fundamentally tests the model’s visual reasoning ability.
>
>
> We also fully agree with the reviewer’s suggestion to more formally define visual reasoning. In future work, we plan to explicitly annotate the types of visual reasoning involved in each of the seven categories tasks of BLINK-Twice. For example, Visual Occlusion entails reasoning over spatial relationships; Physical Illusion involves understanding physical phenomena such as light refraction; and Forced Perspective requires perspective analysis and inference about the camera’s viewpoint. These clarifications will enhance the clarity of our task definition and improve the interpretability of model evaluations.

---

> ### Comment · Reviewer_8pg1 · 2025-08-01
> **Thanks!**
>
> I appreciate the authors writing a long rebuttal. My concerns are addressed for some questions like Q1, Q6, Q7, Q8, Q9, .
>
> However, in the rebuttal, the authors repeatedly say "In future work, we will..."—which is a bit confusing. It’s unclear whether they intend to improve this current paper or introduce a separate follow-up (e.g., BLINK-Twice-V2). From my perspective, it’s more important to strengthen this current version to ensure the benchmark is as useful as possible to the community. It would be helpful if the authors clarified how they plan to improve this specific release.
>
> For Q2, I appreciate the authors introducing three levels of hierarchy, but I think more experiments are needed to justify these arguments.
>
> For Q3, I think the size of the dataset (400) is still a big concern, especially after taking down licensed images.
>
> For Q4, I understand now that the reasoning traces are generated by GPT, then revised by humans. I would think it could be highly impactful to release the model-generated traces to compare against human-verified versions.
>
> For Q10, I would appreciate full disclosure of all images used in the benchmark (especially given the limited size of 400). And whether you plan to update the numbers after removing licensed images.
>
> For Q12, I think what authors suggest is that visual perception is part of visual reasoning?

---

> > ### Author Response · Authors · 2025-08-01
> > **Appreciation for Constructive Feedback**
> >
> > Thank you very much for your thoughtful follow-up and for your engagement with our work. We truly appreciate your constructive suggestions, which are highly valuable for improving this paper.
> >
> > > I would like the authors to clarify whether the proposed future improvements....
> >
> > We clarify that “future work” refers to improvements to this current version of BLINK-Twice, not a separate release (e.g., BLINK-Twice-V2). In fact, several of the items we discussed in the rebuttal—such as dataset refinements (e.g., image replacements to address licensing issues), updates to the Hugging Face release, and improvements to figures and explanations in the paper—are already in progress and will be included in the updated version of BLINK-Twice. We will ensure timely release of updated versions.
> >
> > ---
> >
> > >Q2. More experiments are needed to justify these arguments...
> >
> > We appreciate your recognition of our proposed three-level framework  and agree that further empirical support is important, especially given the novelty of this direction. As noted in the caption of Fig. 6 (Supplementary), our three-leve hierarchy was partly inspired by [11], which offers a detailed taxonomy and analysis of multimodal reasoning, including relevant sections like "Taxonomy of Multimodal Reasoning" and "Active Visual Perception".
> >
> > We fully agree that more experiments are needed to validate this framework. Our main paper already provides empirical evidence for Level 2 (Re-perception) via multi-turn dialogue that enforces repeated image presentation. In this revision, we add new results supporting Level 3 (Collaborative Perception). Specifically, we evaluate OpenAI’s o3 model (API Version)—representative of Level 3 capabilities—and observe improved performance over GPT-4o and o1 (Level 1 One-Pass Perception). Moreover, when we prepend prompts with “Do not use or simulate any visual tools or actions,” —this led to a performance drop, suggesting that models benefitting from visual actions (e.g., zooming or rotation) indeed gain an advantage through active perceptual mechanisms. We hope these additions help substantiate our framework, and we thank you again for encouraging us to clarify and strengthen this aspect of the work.
> >
> > |Model|GPT-4o|o1|o3|o3-no-tools|
> > |-|-|-|-|-|
> > |Overall-Acc|0.571|0.608|**0.633**|0.621|
> >
> >
> > [11] Lin Z, Gao Y, Zhao X, et al. Mind with eyes: from language reasoning to multimodal reasoning[J]. arXiv preprint arXiv:2503.18071, 2025.
> >
> > ---
> >
> > >For Q10, I would appreciate full disclosure of the benchmark images and clarification on whether results will be updated after removing licensed ones.
> >
> > >For Q3, I think the size of the dataset (400) is still a big concern, especially after taking down licensed images.
> >
> > Thank you for raising this important concern regarding dataset size and transparency. For Q10, we have removed a few licensed images (e.g., from Getty) and collected 50+ new license-compliant images to replace them. While we cannot update the Hugging Face release during the review phase, both the dataset and evaluation metrics will be updated immediately after the review.
> >
> > For Q3, we acknowledge the concern about dataset size. The updated BLINK-Twice will still include at least 400 verified images. In addition, thanks to the diverse question formulations and the naturally adversarial samples generated via image editing, the final set of VQA questions will exceed 1,000. We hope this addresses your concerns regarding both the dataset size and the completeness of image disclosure. Thank you again for your constructive suggestions.
> >
> > ---
> >
> > >For Q4, Release the model-generated traces to compare against human-verified versions.
> >
> > Thank you for the suggestion—we fully agree. Releasing both the raw GPT-generated reasoning traces and the human-verified versions would be highly valuable for the community. It can help reveal common logical flaws, hallucinations, or reasoning failures in the GPT outputs. We plan to include both versions in the Hugging Face release after the review process.
> >
> > ---
> >
> > >For Q12, Visual perception is part of visual reasoning?
> >
> > Thank you for the insightful question—we agree that visual perception and visual reasoning are tightly intertwined in our benchmark.
> >
> > In BLINK-Twice, many tasks ultimately require a perceptual judgment—for example, “Is the car floating?” However, such judgments cannot be made through perception alone. The model must engage in visual reasoning steps: checking spatial alignment, analyzing contact points, interpreting shadow cues, and detecting inconsistencies in the visual scene. These processes involve multi-cue integration, physical plausibility reasoning, and logical inference—hallmarks of visual reasoning.
> >
> > From this perspective, our benchmark tasks begin with perception, proceed through reasoning, and arrive at perception-based conclusions. In this flow, perception serves both as input and as the final target, while reasoning is the essential bridge between them.

---

> > > ### Comment · Reviewer_8pg1 · 2025-08-02
> > >
> > > Some concerns after reading authors' response:
> > >
> > > > **Specifically, we evaluate OpenAI’s o3 model (API Version)—representative of Level 3 capabilities—and observe improved performance over GPT-4o and o1 (Level 1 One-Pass Perception). Moreover, when we prepend prompts with “Do not use or simulate any visual tools or actions,” —this led to a performance drop, suggesting that models benefitting from visual actions (e.g., zooming or rotation) indeed gain an advantage through active perceptual mechanisms.**
> > >
> > > 1. Though GPT-o3's web version supports visual zoom in, its API version does not.
> > > 2. The reported performance drop—from 0.633 to 0.621—is minimal?
> > >
> > > These observations leave me unconvinced by the argument. I would suggest removing this claim unless further evidence can be provided.
> > >
> > > > **Thank you for raising this important concern regarding dataset size and transparency. For Q10, we have removed a few licensed images (e.g., from Getty) and collected 50+ new license-compliant images to replace them.**
> > >
> > > Could the authors clarify what is meant by "license-compliant"? I would appreciate a detailed breakdown of the licensing terms for the 400 (including the updated) images.
> > >
> > > > **Thank you for the suggestion—we fully agree. Releasing both the raw GPT-generated reasoning traces and the human-verified versions would be highly valuable for the community. It can help reveal common logical flaws, hallucinations, or reasoning failures in the GPT outputs. We plan to include both versions in the Hugging Face release after the review process.**
> > >
> > > Do the authors have concrete examples of such logical flaws, hallucinations, or reasoning failures, along with how humans corrected them? What proportion of the traces contain such issues? On average, how many words or edits are required per flawed trace to reach the final version?
> > >
> > > As much as I dislike the rebuttal policy this year, I find it difficult to update my evaluation based solely on promises—without actually seeing the updated dataset and newly proposed materials. I hope the ACs will consider a better workaround given these constraints.

---

> > > > ### Author Response · Authors · 2025-08-03
> > > > **Appreciation for Constructive Feedback**
> > > >
> > > > **1.** Thank you for raising this important point. We acknowledge the concern and would like to clarify our experimental assumptions and current limitations.
> > > >
> > > > **First**,  the o3 API can support visual actions (e.g., zoom, crop, rotate) through OpenAI’s Code Interpreter tool, as officially documented:
> > > >
> > > > >“The Code Interpreter tool allows models to write and run Python code in a sandboxed environment to solve complex problems... Use it for: [...] Boosting visual intelligence in our latest reasoning models (like o3 and o4-mini). The model can use this tool to crop, zoom, rotate, and otherwise process and transform images.”
> > > >  — OpenAI Documentation, docs/guides/tools-code-interpreter
> > > >
> > > > Despite this capability being theoretically available, we find that API-level access does not expose intermediate tool calls or their outputs, particularly in image-based reasoning. While for text-only inputs we can detect tool invocation via structured responses such as ResponseCodeInterpreterToolCall, this is not the case for image inputs. We cannot determine whether the model actually invoked zooming or cropping, nor access the intermediate reasoning steps or image transformations. In contrast, the web interface does visibly display such steps. We suspect this limitation may be intentional, likely for security and privacy reasons—e.g., to prevent the extraction of model internals or reasoning traces that could facilitate unauthorized distillation.
> > > >
> > > > **Second**, we agree that the observed performance drop (from 0.633 to 0.621) is modest and does not provide conclusive evidence of visual actions improving performance. While we initially conducted these experiments to further investigate o3’s visual reasoning behavior as part of our rebuttal analysis, in light of these two limitations—the inability to verify tool usage at inference time, and the lack of significant performance difference—we have decided not to include these additional experiments and associated claims in the final version of the paper. We appreciate the reviewer’s critical observation, which helped us clarify the limits of our current findings.
> > > >
> > > >
> > > >
> > > >
> > > > **2.** Thank you for your thoughtful question regarding licensing transparency. By “license-compliant,” we mean that all images used in our dataset are governed by licenses that explicitly allow free use for non-commercial purposes, such as academic research, model evaluation. For example, some images are sourced from Pixabay, which provides content under its own Content License:
> > > >
> > > > >Subject to the Prohibited Uses (see below), the Content License allows users to:
> > > >  ✓ Use Content for free
> > > >  ✓ Use Content without having to attribute the author.
> > > > ✓ Modify or adapt Content into new works
> > > > ...
> > > >  — Pixabay Content License
> > > >
> > > > Our dataset also includes images subject to the Pixabay Creative Commons Zero (CC0) license:
> > > >
> > > > >“This means that to the greatest extent permitted by applicable law, the authors of that work have dedicated the work to the public domain by waiving all of his or her rights... The CC0 Content can be used for all personal and commercial purposes without attributing the author.” — Pixabay Terms of Service, Section on CC0 Content
> > > >
> > > >
> > > >
> > > > This ensures that such CC0 images may be freely used, modified, and redistributed for any purpose, without restriction. This fully satisfies academic and open-access requirements. We have removed a few images with incompatible licenses (e.g., Getty) and replaced them with over 50 newly verified ones.  To further clarify, we will release the dataset under a custom research-only license, explicitly stating that all images are for academic use, model evaluation, and reproducibility studies only. This ensures alignment with the original image licensing terms (e.g., Pixabay, CC0). While assembling such a dataset under strict licensing constraints is not trivial, we have made every effort to ensure that all included images are permitted for free use in non-commercial, academic contexts.

---

> > > > > ### Author Response · Authors · 2025-08-03
> > > > > **Appreciation for Constructive Feedback**
> > > > >
> > > > > **3.** Thank you for your thoughtful follow-up. We agree that concrete examples and quantification are essential for evaluating the quality of the GPT-generated reasoning traces.
> > > > >
> > > > > Example 1: Correction Needed (Image: 44.jpg)
> > > > > - Original GPT output:
> > > > > >Detail clues: The red cloth is draped and hanging over the balcony railing, and its folds and contours give it an impression of a human shape.
> > > > > The truth and cause of illusion: There is no person wearing red clothing. The red cloth, due to its positioning and shape, creates the illusion of a person standing in front of the person on the balcony.
> > > > >
> > > > > - Human-corrected version:
> > > > > >Detail clues: The red shape lacks visible human features such as hands or feet and appears to be held at one end by the woman.
> > > > > The truth and cause of illusion: The red shape is actually a sheet that the woman is shaking out, creating the illusion of a person in red clothing due to its form and position.
> > > > >
> > > > > While the model correctly inferred that the red shape was not a person, it misidentified its position—assuming it was hanging—rather than being actively held by the woman. It also missed visual cues like the absence of hands or feet. The correction improves both spatial grounding and explanatory clarity.
> > > > >
> > > > > Example 2: No Correction Needed (Image: 168.jpg)：
> > > > >
> > > > > >Detail clues: The white area has a fluffy texture indicative of fur, and there is a visible ear at the top of the shape.
> > > > > The truth and cause of illusion: The image is of a white cat curled up with a citrus fruit on its back, resembling the shape and color pattern of a fried egg.
> > > > >
> > > > > Here, the model gave a precise explanation based on fine-grained visual features. No correction was needed. This high quality stems from the fact that the ground-truth identity (“this is a cat not a fried egg.”) was clearly provided in the prompt.
> > > > >
> > > > > Across all reasoning traces, fewer than 10% required correction, and most of these could be addressed with a single round of human edits. GPT’s role in our setup is not to infer from scratch, but to identify supporting visual evidence under a given ground-truth identity. While occasional misinterpretations occu, the majority of traces remain accurate and informative.

---

> > > > > > ### Comment · Reviewer_8pg1 · 2025-08-03
> > > > > >
> > > > > > I appreciate the authors’ response to Q1 and acknowledge that further experiments are needed to fully support the arguments made in the rebuttal/paper.
> > > > > >
> > > > > > I’d like to clarify my original Q2: **I would appreciate a detailed breakdown of the licensing terms for all 400 images, including the newly added ones.**
> > > > > >
> > > > > > My point is: A summary table or statistics would be helpful (if available).
> > > > > >
> > > > > > I also appreciate the clarification to Q3 that only 10% of traces required human correction. My follow-up question is: Could the authors specify the **exact input given to GPT**, ideally by sharing the full prompt used? This would help me better understand the setup.
> > > > > >
> > > > > > Finally, just a quick reminder that my original question Q3 was:
> > > > > > *"Do the authors have concrete examples of such logical flaws, hallucinations, or reasoning failures, along with how humans corrected them?"*
> > > > > >
> > > > > > So just to confirm — the example you provided appears to be a case of **hallucination**, correct? If so, do you also have examples of **logical flaws** or **reasoning failures**, and how those were corrected?

---

> > ### Author Response · Authors · 2025-08-04
> > **Response to Reviewer Comments**
> >
> > **1.** Thank you for your thoughtful follow-up. We appreciate your emphasis on licensing transparency and agree that a detailed breakdown improves clarity. Our dataset includes a diverse mix of freely usable images drawn from well-established platforms. Below is an approximate composition of the 400 images based on source:
> > - **Pexels** : 50–100 newly added images (~25%). All newly added images were sourced from Pexels, which provides free, high-quality content under a permissive license. Pexels allows free use, modification, and redistribution without attribution:
> > “All photos and videos on Pexels can be downloaded and used for free. Attribution is not required. You can modify the photos and videos from Pexels. Be creative and edit them as you like.”
> >
> > - **Pixabay** : Over 100 images (~40%), under either the Pixabay Content License or CC0.
> > - **Unsplash** : ~10% of the dataset. Unsplash permits free use for both commercial and non-commercial purposes, with no attribution required.
> >
> > - **Gratisography** : ~5–10% of the dataset. Gratisography images are free to use and modify for any purpose, including commercial use.
> >
> > - **Others**: A small remainder from additional verified free-use sources.
> >
> > During dataset curation, some initial images obtained from forum-like sources were matched and replaced with identical versions available on publicly verified platforms (e.g., Pexels, Unsplash, Gratisography), each of which provides clear, permissive licensing. This ensures consistency and licensing clarity across the dataset.
> >
> > A more detailed summary of image sources and license types will be included in the supplementary materials. This will include aggregate statistics and representative examples that reflect the dataset composition and licensing structure, ensuring transparency and responsible reuse. We hope this addresses your request, and we sincerely appreciate your close and constructive feedback.
> >
> >
> > **2.** Thank you for your follow-up. Below is the full prompt we used to elicit GPT's generation of both the key visual clues and the explanation of the truth behind the illusion:
> >
> > `````
> > You will be given an image, a caption describing a visual illusion, and a specific question related to the image.
> > Your task is to analyze this input through **a five-step reasoning chain**, and then generate **two key summary points**.
> >
> > Please follow these five reasoning stages:
> >
> > 1. Initial perception : Describe the first impression based on the caption provided, considering the question. What does the viewer initially think when they look at the image?
> > 2. Identifying misleading : Explain which elements or characteristics in the image appear unusual, odd, or different from what one would expect. Focus on features that might lead to confusion or a mistaken interpretation in relation to the question.
> > 3. Detailed visual clues: Provide the steps that lead to understanding the actual cause of the illusion, using details from the caption. Explain how a deeper look helps to clarify the situation and the misunderstanding.
> > 4. True Reality : Explain how the initial misinterpretation is challenged and corrected in relation to the question.
> > 5. Final answer : Summarize the correct understanding of the image after reasoning through the illusion and provide the final answer to the question.
> >
> > After completing the above reasoning process, summarize the conclusion in the following format:
> > 1. **Detail clues**: [What are the directly visible features in the image (e.g., shape, alignment, shadows) that support the correct answer? Do not explain why the illusion works — focus only on what is visually present.]
> > 2. **The truth and cause of illusion**: [What is really happening in the scene, and what causes the misperception? Explain in 1–2 concise sentences.]
> >
> > Each input includes:
> > - **Image**: A visual illusion scene
> > - Caption (`generated_caption`): A brief description of the image and illusion
> > - **Question**: A multiple-choice question about the image
> >
> > Make your reasoning and final output:
> > - Step-by-step, precise, and concise
> > - Focused on visual evidence and logical reasoning
> > - Consistent in tone and structure
> >
> > `````

---

> > ### Author Response · Authors · 2025-08-04
> > **Response to Reviewer Comments**
> >
> > **3.** Thank you for your insightful observations. Regarding your second point: yes, the example we previously shared (44.jpg) primarily illustrates a perceptual hallucination, but it also reveals a logical flaw. Specifically, the model did not infer that the red figure was a piece of cloth based on the absence of human features (such as hands or feet). Instead, it directly asserted that the shape was a red cloth rather than a person, without first establishing the necessary visual evidence. This indicates a lack of reasoning logical rigor, not just a visual illusion. Furthermore, the claim that the cloth is “draped and hanging over the balcony railing” is also inaccurate — in reality, it is being actively held by the woman. This is another hallucinated detail, unrelated to any visible support structure. In other words, a single example requiring revision may simultaneously contain both hallucinated content and logical flaws in its reasoning.
> >
> > In addition, we also include below a case of reasoning failure:
> >
> > - Corrected human version:
> >
> > >Detail clues: The visible hands are human and positioned on the steering wheel, while the dog’s body is in front, obscuring the rest of the driver.
> > The truth and cause of illusion: The dog is sitting in a way that hides the driver’s body, creating the illusion that the dog has human hands and is driving the car.
> >
> > - Original GPT output:
> >
> > >Detail clues: The dog is facing forward and its body is aligned with the steering wheel, making it appear natural for it to be driving.
> > The truth and cause of illusion: The dog has learned to place its paws on the wheel, mimicking human behavior, which creates the illusion of it driving.
> >
> >
> > In this case, the model’s output reflects a clear reasoning failure. Although the provided caption already indicated that the dog is not actually driving, GPT failed to identify the key visual clue — namely, the presence of human hands on the steering wheel. As a result, it constructed a seemingly plausible but incorrect explanation, rather than reasoning its way to the correct interpretation. Similarly, while this example also involves hallucination, it exhibits a more clear-cut reasoning error compared to the earlier “red” case. We hope these clarifications help illustrate multiple types of model errors. Thank you again for your careful review and thoughtful feedback.

---

> > > ### Comment · Reviewer_8pg1 · 2025-08-04
> > >
> > > I appreciate the detailed reports on licensing terms and agree that the authors should release them in the supplemental.
> > >
> > > For Q2, I was specifically asking for **the exact input provided to GPT**, so it would be helpful to share the actual “Caption (generated_caption)” field for those two examples (or more). Feel free to reference images on your Hugging Face page—I’ll take a look there.
> > >
> > > Also, why is it called generated_caption? How is it generated?

---

> > > > ### Author Response · Authors · 2025-08-04
> > > > **Response to Reviewer Comments**
> > > >
> > > > Thank you for the positive and constructive comment.
> > > >
> > > > The "Caption (generated_caption)" refers to the human-generated initial prompt that we manually annotated for each image.
> > > >
> > > > For example, as mentioned earlier:
> > > >
> > > > - 44.jpg: “The image appears to show a woman talking to a person in red behind a railing, but in fact, there is only one woman dressed in black.”
> > > >
> > > > - 168.jpg: “The image looks like a fried egg at first glance, but it’s actually a cat.”
> > > >
> > > > And from our Hugging Face page（It is the images folder instead of image-paired）:
> > > >
> > > > - 1.jpg: “The image appears to depict a statue wearing glasses, but in fact, is a traffic light.”
> > > >
> > > > - 3.jpg: “The kitten appears to have human-like arms, but it’s actually wrapped in a towel.”
> > > >
> > > > These captions serve as concise summaries of the visual misinterpretation and help GPT-4o generate more accurate and logically grounded reasoning traces. Feel free to explore additional examples on our Hugging Face page. Let us know if any further clarification would be helpful.

---

### Official Review · Reviewer_VEfB · 2025-06-27

**Rating:** 5
**Confidence:** 4

**Summary:**

The paper introduces BLINK-Twice, a new benchmark designed to evaluate the visual reasoning abilities of multimodal large language models (MLLMs). Unlike existing benchmarks that rely heavily on language-based reasoning or shallow visual tasks, BLINK-Twice focuses on image-grounded reasoning, requiring models to engage in fine-grained observation and analysis of complex visual content. The benchmark includes 345 carefully selected images spanning seven types of visual challenges, 103 natural adversarial image pairs, and over 1,700 annotated reasoning steps. The authors evaluate 20 leading MLLMs, both open- and closed-source, using metrics including CoT-Score (Chain-of-Thought) to assess reasoning depth and accuracy.

**Dataset Code Accessibility:**

Partly

**Dataset Code Comments:**

The dataset and corresponding code are provided.

**Ethical Considerations:**

No, there are no or only very minor ethics concerns

**Final Justification:**

The authors’ rebuttal has satisfactorily addressed my primary concerns regarding the evaluation methodology and experimental completeness. Specifically:
- The additional experiments and analyses provided during the rebuttal period effectively resolved my concerns about the benchmark’s evaluation depth and its applicability to existing models.
- The authors’ detailed discussion on how their benchmark can inspire future improvements in reasoning architectures, visual interaction mechanisms, and process-level supervision further demonstrates the significance and forward-looking value of this work.

Given the overall quality of the paper, its clear contributions to multimodal reasoning evaluation, and the thoughtful responses during the rebuttal, I am inclined to recommend acceptance.

**Limitations Weaknesses:**

1. The benchmark primarily focuses on visually challenging phenomena including optical illusions, camouflage and forced perspective. While this reveals reasoning deficits in MLLMs, it may also introduce a bias toward rare or atypical image types, potentially limiting the generalizability of results to real-world scenarios where reasoning must occur on more mundane or diverse visual content.
2. The adopted metrics including CoT-Score capture different facets of model behavior, but they lack unified calibration or benchmarking against human baselines, making it difficult to interpret the absolute quality of reasoning. Additionally, these metrics may be sensitive to verbosity or stylistic differences in model output, which is not necessarily reflective of reasoning quality.
3. The benchmark tests static visual reasoning in single images or adversarial image pairs. However, temporal reasoning (e.g., across video frames or causal sequences) and cross-modal interactions are not addressed, despite being central to many real-world perceptual tasks.

**Strengths Contributions:**

1. The authors significantly advance the field by introducing a benchmark that shifts focus from surface-level perception to deep visual reasoning, bridging a crucial gap in existing evaluation methods.
2. The dataset comprises seven categories of visual phenomena (e.g., forced perspective, motion illusion), which are both naturalistic and semantically rich, increasing ecological validity. The method of creating natural adversarial image pairs using GPT-4o’s editing capabilities is novel and effective in reducing reliance on commonsense priors.
3. Detailed step-by-step reasoning chains are included to allow for fine-grained assessment beyond final-answer accuracy. The benchmark is rigorously validated on 20 MLLMs under zero-shot conditions, offering detailed insights into the performance gap between visual perception and reasoning.

---

> ### Author Rebuttal · Authors · 2025-07-30
>
> **Q1.** Thank you for the reviewer’s insightful comment. We agree that our benchmark focuses on visually challenging phenomena such as optical illusions, camouflage, and forced perspective, which may not fully represent the diversity of real-world visual reasoning scenarios.
>
> However, this was a deliberate design choice. In everyday images, MLLMs often rely on superficial visual cues and perform reasoning primarily within the language space, without engaging in genuine visual inference. This behavior has been extensively examined in prior work. In contrast, our benchmark deliberately includes counter-intuitive and challenging cases to break such shortcuts, thereby providing a stronger test of a model’s ability to truly reason over visual content. For example, visual occlusion tasks assess spatial understanding, while forced perspective challenges the model’s grasp of depth cues and camera viewpoints. While the images in our benchmark are relatively rare, the reasoning skills they test—such as spatial inference, compositional understanding, and physical plausibility—are fundamental and broadly applicable to many real-world tasks. Moreover, targeting a specific domain to test a model’s ability is a common and well-established strategy in benchmark design.
>
> We fully acknowledge the importance of also covering more everyday scenarios. In future work, we plan to expand our benchmark to include more naturalistic visual reasoning tasks, such as causal inference and action prediction, to enhance the comprehensiveness and generalizability of the evaluation.
>
> ---
>
> **Q2.** We appreciate the reviewer’s thoughtful feedback. In response to concerns about the lack of human baselines and potential stylistic bias in CoT-Score, we provide targeted analyses below to validate its reliability and interpretability.
>
> ***(1) Human baselines:*** We fully agree that aligning model performance with human capabilities is crucial for evaluating the absolute level of reasoning quality. However, we observed that, after completing several examples, participants often began to sense the presence of "tricks" embedded in the task, which led them to adopt a state of heightened alertness and engage in unusually careful and detailed analysis. This strategic shift differs from the more intuitive judgments typically made by users in natural settings.To establish a more reasonable human reference baseline, we designed two complementary evaluation protocols: (1) Full Evaluation: Two users completed the entire test set, representing the performance of experienced human users; (2) Limited Sampling Evaluation: 10 new users each answered 5 randomly selected samples, simulating first-time, natural judgments typical of ordinary users.
>
> ||Human-5|Human-all|InternVL2-40B|Qwen2.5-VL-72B|GPT-4o|Gemini-2.5-pro|
> |-|-|-|-|-|-|-|
> |CoT Score|0.580|0.847|0.301|0.360|0.601|0.584|
>
>
> Experimental results show that baseline models such as InternVL2-40B and Qwen-2.5VL perform worse than ordinary human users. While stronger MLLMs like GPT-4o and Gemini-2.5-pro achieve performance comparable to ordinary users, there remains a clear gap between these models and experienced human performance.
>
> ***(2) CoT-Score Metrics:*** Regarding the second concern, we agree with the reviewer that existing metrics may be affected by stylistic or verbose outputs, potentially reducing their validity in reasoning evaluation. To address this, CoT-Score avoids text-similarity-based metrics (e.g., BLEU, BERTScore), which are overly sensitive to surface features and fail to reflect actual reasoning. Instead, CoT-Score adopts a GPT-4o-based multimodal scoring method, focusing on whether the model identifies key visual evidence and logically derives conclusions aligned with the image truth, thus reducing bias from linguistic style. In addition, the scoring criteria focus only on the key reasoning steps and essential content, allowing for diverse reasoning paths and expressions.
>
> To further assess the metric’s robustness to language style, we conducted a paraphrasing experiment: we selected two full-score answers generated by GPT-4o and rephrased them using GPT-4o itself—preserving the semantic content and reasoning structure, while altering wording and sentence order. Re-evaluation with CoT-Score showed only slight variations, with scores remaining largely consistent. This supports the claim that CoT-Score emphasizes content coverage and reasoning coherence rather than superficial linguistic form.
>
> | Model              | Original-Score | Rewrite-Score |
> |--------------------|----------------|----------------|
> | InternVL2_5-8B     | 0.287          | 0.303          |
> | QVQ-72B            | 0.438          | 0.439          |
> | Gemini-2.0-flash   | 0.469          | 0.482          |
> | Claude-3.7         | 0.526          | 0.516          |
>
>
> In addition, we conducted a consistency study between CoT-Score and human judgments. We sampled 200 responses from multiple models, and three human annotators independently scored each. The results show strong agreement between CoT-Score and human ratings, supporting its reliability for reasoning evaluation. Notably, recent work such as MME-CoT (ICML 2025) has also adopted similar GPT-based scoring methods, reflecting the growing acceptance of such approaches in multimodal reasoning tasks.
>
> | Model              | Human alignment |
> |--------------------|------------------|
> | QVQ-72B            | 0.86             |
> | Claude-3.7         | 0.94             |
> | GPT-4o             | 0.92             |
> | Gemini-2.0-flash   | 0.92             |
>
> ---
>
> **Q3.** We appreciate the reviewer’s insightful comment and fully agree that temporal reasoning and cross-modal interactions are central to many real-world perceptual tasks. As acknowledged in our supplementary material (Section C, Lines 146–149), we have highlighted the importance of extending BLINK-Twice to support video-based reasoning. Our current work offers an initial exploration of static image-based reasoning, laying a foundation for future extensions into dynamic and cross-modal settings. We view such extensions not only as important, but also as a natural next step—one that we are actively considering in the continued development of the benchmark.

---

> > ### Comment · Area_Chair_3BtT · 2025-08-05
> >
> > Dear Reviewer,
> >
> > Please engage in the rebuttal discussion and indicate whether the authors' response addresses your concerns. Thank you for your time and effort.
> >
> > Best,
> > Area Chair

---

> > ### Comment · Reviewer_VEfB · 2025-08-05
> >
> > Thank you for the detailed response and the additional experiments provided, which have effectively addressed my primary concerns. I would like to further inquire: since your work mainly contributes an evaluation benchmark, do you have any insights or suggestions on how this benchmark could inspire future improvements in the reasoning capabilities of open-source models?

---

> ### Author Response · Authors · 2025-08-06
>
> Thank you very much for your thoughtful follow-up. We are glad to hear that our previous response and experiments have addressed your main concerns. Regarding your question on how BLINK-Twice may inspire future improvements, we offer the following perspectives:
>
> - **Enhancing Visual Reasoning Architectures.** Evaluation results on BLINK-Twice reveal that current multimodal large language models still struggle with fine-grained visual reasoning, often relying heavily on language priors rather than truly grounded visual understanding. This highlights the need for future model architectures to better support perception-centered reasoning. We hope that benchmarks like BLINK-Twice can help motivate the development of models with improved iterative visual attention, spatial grounding, and step-by-step visual reasoning—moving beyond shallow perception and toward more robust, visually grounded inference.
>
> - **Tool-Use and Visual Interaction.** The performance gains observed in models like o3 (web) with Code Interpreter suggest that visual interaction tools (e.g., cropping, zooming, rotating) and repeated image observation can substantially improve reasoning. This points toward interactive visual reasoning as a promising future direction, where open-source models may benefit from incorporating similar capabilities or training signals. We hope BLINK-Twice can help steer open-source models toward incorporating visual interaction mechanisms as part of their reasoning toolkit.
>
> - **Supervising the Reasoning Proces.** BLINK-Twice reveals that current models often produce flawed or incomplete reasoning chains, even when the final answer happens to be correct. This gap underscores the need for training approaches that move beyond final-answer supervision, encouraging models to generate more accurate and interpretable reasoning steps. We believe our benchmark’s focus on visual reasoning chains can inspire future work on incorporating intermediate-process supervision into model training, potentially leading to more stable and transparent inference.
>
>
> We are currently exploring these directions ourselves and are excited to see how the community leverages BLINK-Twice to push forward the frontier of multimodal reasoning. We sincerely thank you again for your thoughtful feedback and for encouraging us to reflect on the broader impact of our benchmark.

---

> > ### Comment · Reviewer_VEfB · 2025-08-06
> >
> > Thank you for the comprehensive follow-up and for sharing your perspectives on how BLINK-Twice can inspire future advancements. I strongly recommend that you incorporate these valuable discussions and the additional experiments conducted during the rebuttal phase into the revised version of the paper. Given the quality and contributions of this work, I am inclined to further raise my score to support its acceptance.

---

> ### Author Response · Authors · 2025-08-06
>
> Dear Reviewer VEfB,
>
> Thank you very much for your kind and encouraging follow-up. We greatly appreciate your thoughtful comments, as well as your recommendation to include our additional discussions and experiments in the final version of the paper. We will make sure to incorporate these insights into the revised manuscript.
>
> Thank you again for your time and support.
>
> Sincerely,
>
> The Authors

---

### Official Review · Reviewer_e2J2 · 2025-07-03

**Rating:** 6
**Confidence:** 3

**Summary:**

This paper introduces a visual reasoning benchmark and places itself as a more complex analog of the BLINK benchmark. BLINK-Twice includes seven types of visual challenges, adversarial image pairs, and annotated reasoning chains to allow for multiple levels of understanding model performance. The authors also provide evaluations of multiple SOTA open-source and closed-source VLMs.

**Dataset Code Accessibility:**

Yes

**Dataset Code Comments:**

The dataset was accessible with the provided code after removing the "datasets/" prefix.

**Ethical Considerations:**

No, there are no or only very minor ethics concerns

**Final Justification:**

See response comments below.

**Limitations Weaknesses:**

1. I may have missed this point, but it doesn't seem like the paper justifies the process of annotating the reasoning traces and using them as ground truth. While I appreciate the attempt to have more granular evaluations, I worry that leaderboard-climbing on this measure will lead to models with very particular types of reasoning traces that align with GPT-4o generations. Additionally, it has been found that models' output "reasoning traces" can be superficial and do not necessarily map onto the reasoning that is being done in latent space. Can the authors address this point to better justify the way they measure CoT accuracy here?
2. The dataset is actually relatively small. Can the authors add some error bars so we have a sense of what the meaningful variance is? Knowing the amount of noise to take into account is very important when drawing qualitative conclusions about e.g., rank-ordering of models. Additionally, as LLM-as-a-judge evaluations can also be highly variable, error bars here would be very helpful as well.
3. Figure 2 feels unnecessarily crowded. While the intended messaging is useful, the icons and flowchart that are not part of a specific real example (in parts b and c) are too vague to really understand what that step means. I would recommend picking a real example illustrative of the point to maximize clarity.

**Strengths Contributions:**

This benchmark offers well-rounded coverage of different visual abilities, and "visual reasoning" for the scope of this paper is clearly defined. The figures are clean with illustrative examples for the different aspects of BLINK-Twice. Figure 3 is an especially helpful snapshot, and is also easy to understand (however, I have some some concerns about Figure 2 stated below). The evaluations are thorough, and CoT score is a nice touch beyond high-level accuracy (though I have some questions/concerns about it, stated below). Additionally, the paper is well-written and easy to follow.

---

> ### Author Rebuttal · Authors · 2025-07-30
>
> **Q1.**  We appreciate the reviewer’s thoughtful feedback and agree that evaluating the reasoning traces of MLLMs requires careful justification. Below, we clarify the annotation process of our reasoning traces and the rationale behind the design of CoT-Score, along with multiple experiments we conducted to validate its reliability and robustness.
>
> ***(1) Verified Reasoning Paths:*** Our reasoning path annotations are not solely generated by GPT-4o, but rather produced through a hybrid process involving human annotators and MLLM-assisted generation (Lines 164–176). We start from human-written factual image descriptions (e.g., "This image appears to depict a cat and its shadow, but in fact shows a yellow cat and a black cat") and guide GPT-4o to generate a structured five-stage reasoning chain, thereby mitigating hallucination-induced errors. Moreover, all generated reasoning paths undergo rigorous human verification: each GPT-4o-produced chain is independently reviewed at least twice by a team of five annotators. This process ensures that our annotated reasoning paths are both accurate and suitable as evaluation references.
>
> ***(2) CoT-Score Robustness:*** We fully acknowledge the reviewer’s concern that leaderboard-driven optimization on this metric may lead models to generate reasoning traces that closely align with GPT-4o-style outputs, without necessarily reflecting genuine improvements in reasoning ability. To address this, CoT-Score deliberately avoids surface-level similarity metrics such as BLEU or BERTScore. Instead, it adopts a GPT-based semantic evaluation approach, focusing on whether the model identifies key visual evidence and reveals the ground truth of the image, rather than matching any particular language style. In addition, the scoring criteria focus only on the key reasoning steps and essential content, allowing for diverse reasoning paths and expressions.To further assess robustness against stylistic bias, we conducted a rewriting robustness experiment. We selected two high-scoring GPT-4o responses and prompted GPT-4o to rewrite them in a different style while preserving semantic content. We then re-evaluated the CoT-Score and observed a slight but consistent increase in scores, while overall stability was maintained. This suggests that CoT-Score is more sensitive to the coverage of essential content rather than superficial stylistic conformity.
>
>
> | Model              | Original-Score | Rewrite-Score |
> |--------------------|----------------|----------------|
> | InternVL-2.5-8B     | 0.287          | 0.303          |
> | QVQ-72B            | 0.438          | 0.439          |
> | Gemini-2.0-flash   | 0.469          | 0.482          |
> | Claude-3.7         | 0.526          | 0.516          |
>
>
>
> ***(3) Human–GPT Agreement on CoT-Score:*** To assess the reliability of CoT-Score, we conducted a consistency study comparing GPT-based scoring with human judgments. We randomly sampled 50 questions from each of four models (200 reasoning responses in total), and asked a human raters to independently assign scores. If the GPT and human scores were the same (e.g., both 0 or both 1), we marked it as an alignment. We then calculated the overall alignment rate between GPT and human scoring. The results indicate a high level of agreement between human and GPT scores, validating the reliability of CoT-Score as a reasoning evaluation metric.
>
>
> | Model              | Human alignment |
> |--------------------|------------------|
> | QVQ-72B            | 0.86             |
> | Claude-3.7         | 0.94             |
> | GPT-4o             | 0.92             |
> | Gemini-2.0-flash   | 0.92             |
>
> ***(4) Explicit Output-Based Reasoning Evaluation:*** We also agree with the reviewer that evaluating only the reasoning traces generated by MLLMs may not fully reveal the underlying latent reasoning processes. Future work could explore deeper mechanisms such as attention trajectories or neural activations to better understand the internal reasoning dynamics of large models. However, such approaches typically require access to the model’s internal architecture and intermediate states, making them infeasible for closed-source models and often non-trivial to generalize across different open-source architectures due to compatibility and interpretability issues. In contrast, analyzing the explicit reasoning traces output by models offers a more universally applicable, comparable, and interpretable method for assessing reasoning capabilities, especially as a complementary perspective to traditional accuracy metrics. Recent works such as MINT-CoT and MME-CoT (ICML 2025) have also adopted similar output-based evaluation paradigms, further supporting the validity and emerging trend of this approach.
>
> ---
>
> **Q2.** We appreciate the reviewer’s attention to the stability and ranking reliability of model evaluation. We fully agree that, especially for relatively small datasets, reporting uncertainty is crucial when qualitatively comparing model performance. In response, we applied 1000-round bootstrap resampling to compute the mean and standard deviation of each model’s overall accuracy, and have added corresponding error bar analyses to the paper. Bootstrap resampling estimates the standard deviation by repeatedly sampling with replacement from the data and computing the standard deviation of the resulting accuracies. Below are results for several representative models.
>
>
> | Model            | Overall Accuracy (mean ± std) | CoT Score (mean ± std)   |
> |------------------|-------------------------------|---------------------------|
> | QVQ-72B          | 0.5754 ± 0.0187               | 0.4378 ± 0.0279           |
> | Qwen2.5-VL-32B   | 0.5754 ± 0.0186               | 0.3273 ± 0.0274           |
> | GPT-4o           | 0.5641 ± 0.0187               | 0.6059 ± 0.0282           |
> | O1               | 0.6088 ± 0.0189               | -                         |
> | Gemini-2.5-Pro   | 0.6675 ± 0.0185               | 0.5833 ± 0.0419           |
>
>
>
> We observed that despite the limited dataset size, the standard deviations across models remain relatively small (around ±0.018–0.04), indicating good stability, particularly in overall accuracy. Due to rebuttal space constraints, we only report partial error analyses here. In the revised version, we will include the full bootstrap results and error bars for all models, and clearly indicate the uncertainty range of model rankings in the plots. We thank the reviewer again for this constructive suggestion, which has helped us present our results more transparently and rigorously.
>
> ---
>
>
> **Q3.** We appreciate the reviewer’s feedback regarding Figure 2. Due to the requirement of the rebuttal process, we are unfortunately unable to include a revised figure in our response. We provide a detailed textual description below regarding the modifications made to Figure 2.
>
> - Part (b): Simplified Pipeline Visualization. We have simplified the content in part (b) to better focus on the core logic of the pipeline. Specifically, we first rewrite the original VQA samples to produce alternative factual outcomes in text form. Based on these rewritten answers, we generate corresponding text editing instructions, which are then passed to GPT-4o for image editing. This process results in natural adversarial examples. In this revision, we reduced the use of abstract elements and diagrams, and instead emphasized a clearer and more intuitive textual explanation..
>
> - Part (c): Concrete Annotation Example. For part (c), we have replaced the original annotation flowchart with a concrete example illustrating the original image, human annotation, and GPT-4o’s step-by-step CoT reasoning output. This example mirrors the style of Figure 1(c) and makes the annotation process more intuitive and accessible.
>
> We thank the reviewer again for this constructive suggestion to improve clarity.

---

> > ### Comment · Reviewer_e2J2 · 2025-08-04
> >
> > Thank you for the thorough response to my review. All my concerns have been addressed, and I recommend that especially the response to Q1 be included in the paper (at least supplement) to increase credibility of the reasoning traces. I will increase my score to a 6.

---

> > ### Author Response · Authors · 2025-08-05
> >
> > Dear Reviewer e2J2,
> >
> > Thank you very much for your thoughtful review and for taking the time to carefully engage with our work. We greatly appreciate your positive feedback and are glad to hear that your concerns have been addressed.
> >
> > As you suggested, we will incorporate our response to Q1 into the final version of the paper to enhance the credibility of the reasoning traces.
> >
> > Thank you again for your time and expertise.
> >
> > Sincerely,
> >
> > The Authors

---

### Note · Authors · 2025-08-14

Dear Reviewers, Area Chair, and Senior Area Chair:

Following the comprehensive responses and in-depth discussions with all reviewers, we sincerely thank you for your thoughtful feedback and constructive engagement during the review process.

**Review Highlights:**

We are encouraged that the reviewers recognized and valued our key contributions, including:

  - **Benchmark novelty** – The BLINK-Twice benchmark transitions from surface-level perception to deep visual reasoning, offering seven engaging task types that compel MLLMs to rely on visual input (VEfB, 8pg1, iirm).
  - **Adversarial data generation** – A novel GPT-4o-based natural adversarial image-pair generation method that reduces reliance on commonsense priors (VEfB, iirm).
  - **Beyond simple accuracy** – Evaluation incorporates CoT scoring and detailed reasoning chains to assess reasoning quality (e2J2, VEfB, iirm).
  - **Extensive experiments** – Comprehensive coverage of both common MLLMs and reasoning models, with qualitative and quantitative evaluation (e2J2, VEfB, 8pg1).
  - **Clear presentation** – Well-written paper with accessible, informative visualizations (e2J2, 8pg1, iirm).

**Main Concerns Addressed:**

  During the rebuttal, we implemented substantial clarifications and improvements to address reviewers’ main concerns:

  - **Clarified CoT Score** – Emphasizing its focus on key content over language style. Added robustness tests under LLM rewrites and human–GPT scoring consistency analysis.
  - **Expanded evaluation results** – Included variance results for Accuracy and CoT Score; added detailed per-model results across all visual challenge categories.
  - **Detailed reasoning path annotation** – Fully described the annotation pipeline, including original human annotations, GPT-4o prompts, and quality-control corrections, with multiple concrete annotation examples.
  - **Enhanced clarity and presentation** – Improved visuals in Fig. 2 and Fig. 4, clarified dataset copyright and filtering, and explained differences among visual challenge categories.
  - **Future directions** – Discussed how BLINK-Twice can inspire future reasoning model development, such as architecture enhancements, visual tool integration, and fine-grained reasoning supervision.


We are grateful for the time and effort invested by the reviewers, Area Chair, and Senior Area Chair in evaluating our work and facilitating a productive exchange during the discussion period.

Best regards,

The Authors

---

### Decision · Program_Chairs · 2025-09-18

**Decision:**

Accept (poster)

**Comment:**

This paper introduces BLINK-Twice, a vision-centric reasoning benchmark that compels MLLMs to reason directly from visual content rather than relying on language-based cues. Reviewers praised the benchmark’s novelty, experimental rigor, and clear presentation, and in the rebuttal, the authors effectively addressed concerns by clarifying CoT scoring, expanding evaluation results, detailing the annotation pipeline, and improving figures and explanations. With all reviewers maintaining or raising their positive assessments, the AC finds this to be a timely and impactful contribution and therefore recommends acceptance (spotlight).

===== FINAL UPDATE FROM DB Track PCs ====

The final decision for this paper has been taken by the program chairs after consultation with the SACs. All Senior Area Chairs have ranked papers according to the feedback from the AC during the review process. We decided to leave the original meta-review to reflect the opinion of the AC in light of the initial discussions with reviewers and SAC.